# BioBRIDGE: Bridging Biomedical Foundation Models via Knowledge Graphs

**Zifeng Wang** [*]
University of Illinois Urbana-Champaign
zifengw2@illinois.edu

**Zichen Wang**
Amazon AWS AI
zichewan@amazon.com

**Balasubramaniam Srinivasan**
Amazon AWS AI
srbalasu@amazon.com

**Vassilis N. Ioannidis**
Amazon Search
ivasilei@amazon.com

**Huzefa Rangwala** [†]
Amazon AWS AI
rhuzefa@amazon.com

**Rishita Anubhai**
Amazon AWS AI
ranubhai@amazon.com

## Abstract

Foundation models (FMs) learn from large volumes of unlabeled data to demonstrate superior performance across a wide range of tasks. However, FMs developed for biomedical domains have largely remained unimodal, i.e., independently trained and used for tasks on protein sequences alone, small molecule structures alone, or clinical data alone. To overcome this limitation, we present BioBRIDGE, a parameter-efficient learning framework, to bridge independently trained unimodal FMs to establish multimodal behavior. BioBRIDGE achieves it by utilizing Knowledge Graphs (KG) to learn transformations between one unimodal FM and another without fine-tuning any underlying unimodal FMs. Our results demonstrate that BioBRIDGE can beat the best baseline KG embedding methods (on average by $\sim 76.3\%$) in cross-modal retrieval tasks. We also identify BioBRIDGE demonstrates out-of-domain generalization ability by extrapolating to unseen modalities or relations. Additionally, we also show that BioBRIDGE presents itself as a general-purpose retriever that can aid biomedical multimodal question answering as well as enhance the guided generation of novel drugs. [1]

## 1 Introduction

Foundation models (Bommasani et al., 2021) trained on large volumes of data can be leveraged and adapted for different domains. In biomedicine, FMs are trained to ingest text corpora (Gu et al., 2021) from scientific literature, protein data in sequences and 3D-structures (Jumper et al., 2021), molecule in graphs and SMILES strings (Fabian et al., 2020) and protein-interaction data in the form of relational graphs. These pre-trained biomedical FMs have achieved a significant gain in comparison to previous methods trained on smaller datasets (Qiu et al., 2023). Introducing multimodal data in training further boosts the performance of FMs, especially in few-shot/zero-shot prediction settings (Radford et al., 2021). In the biomedical domain, for drug-text (Edwards et al., 2022), protein-text (Liu et al., 2023), and drug-protein data (Huang et al., 2021; Ioannidis et al., 2020), multimodal data was leveraged by the joint optimization of unimodal encoders. However, this idea encounters key issues when scaling beyond two modalities:

**Computational Cost**. These approaches require many unimodal encoders with approximately similar sizes to avoid impeding each other. This setup can cause a size explosion of the bundled models

---

[*]This work was completed while the author was an intern at Amazon.

[†]Huzefa Rangwala is on LOA as a Professor of Computer Science at George Mason University. This paper describes work performed at Amazon.

[1]Code is at https://github.com/RyanWangZf/BioBridge.

in a magnitude of the number of modalities, thus rendering a computational burden for performing joint optimization.

**Data Scarcity**. They require pairwise cross-modal datasets of similar size to ensure stable training. The dataset quantity increases exponentially in a combinatorial order of $\binom{K}{2}$ where $K$ represents the number of modalities, inevitably leading to data scarcity.

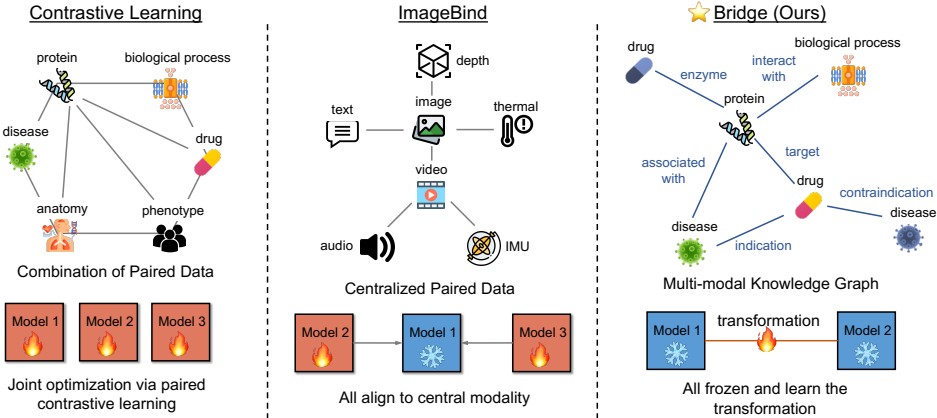

Figure 1: The conceptual comparison between our methods and previous methods. **Left**: multimodal contrastive learning, e.g., CLIP, learns from a combination of paired data, updating all unimodal encoders; **Middle**: ImageBind aligns all modalities with the central modality, with only the central model frozen; **Right**: `BioBRIDGE` learns the transformation across modalities from a multi-modal KG, keeping all FMs frozen.

Unlike ImageBind (Girdhar et al., 2023), which sets the image as the central modality and aligns all the other encoders with the image via fine-tuning, the proposed `BioBRIDGE` keeps all the unimodal FMs fixed and learns to bridge these unimodal FMs. The conceptual demonstration is shown in Figure 1. Specifically, `BioBRIDGE` learns the cross-modality transformation from biomedical knowledge graphs (KGs). This approach is modeled by leveraging the following insights:

**Data Sufficiency**. It is usually easier to collect unimodal data than to collect paired data from two modalities. For instance, close to 250M protein sequences (Rives et al., 2021) and 1.5B molecule structures (Sterling & Irwin, 2015) are available to perform self-supervised pre-training, while only 441K protein-text pairs are one of the largest of biological multimodal datasets (Liu et al., 2023). As such, compared to the joint training of multimodal encoders, bridging independent models that were trained on unimodal data at scale enjoys the merits of data sufficiency and efficiency.

**Structural Transformation**. Multimodal biomedical KG contains the structure information represented by the triplets of head and tail biomedical entities and their relationships. It covers a rich set of modalities such as protein, molecule, and disease (Chandak et al., 2023), which enables comprehensive biomedical analytics and ML. We align the embedding space of unimodal FMs through a cross-modal transformation model utilizing the rich structure in KG triplets.

In summary, `BioBRIDGE` aims to create a universal bridging mechanism capable of efficiently connecting the representations of any pairs of modalities. Technically, the bridge modules are supervised by the rich structure information from knowledge graphs, while the unimodal FMs are kept frozen to advance the parameter and computation efficiency. Experiment shows that:

- The bridged unimodal FMs are competitive in diverse cross-modal prediction tasks.

- `BioBRIDGE` can extrapolate to nodes that are not present in the training KG with comparable performance as the supervised baselines.

- `BioBRIDGE` generalizes to relationships that do not exist in the training KG, and the performance can be enhanced with further training.

## 2    RELATED WORK

Foundation models have sparked remarkable breakthroughs in natural language processing (Brown et al., 2020) and computer vision (Kirillov et al., 2023). In biomedicine, FMs are trained with masked language modeling for text (Gu et al., 2021), proteins (Rives et al., 2021; Lin et al., 2023), drug molecules (Wang et al., 2019a), or with generative modeling for text (Taylor et al., 2022), protein (Madani et al., 2023), molecule (Bagal et al., 2021). They are trained on unimodal data and used as feature extractors in supervised prediction tasks such as protein-protein interaction (Wang et al., 2019b; Hallee & Gleghorn, 2023), protein function prediction (Gligorijević et al., 2021; Wang et al., 2022a) and drug-target interaction (Sledzieski et al., 2022; Kalakoti et al., 2022).

In the literature, multimodal biomedical FMs leverage contrastive learning on the pairs of image-text (Wang et al., 2022b), drug-protein (Huang et al., 2021), drug-text (Edwards et al., 2022) and protein-text (Liu et al., 2023; Xu et al., 2023b;a). Nonetheless, extending these approaches beyond two modalities is challenging due to the need for large volumes of pairs of multimodal datasets. A recent effort (Girdhar et al., 2023) proposed to combine pre-trained models from diverse modalities by fine-tuning all unimodal encoders to be aligned with the image encoder. By contrast, BioBRIDGE does not involve the base FMs in training but trains the transformation bridge modules. In addition, the rich structure information in KG is leveraged to relax the central modality assumption and allows for a more controlled cross-modality transformation. We have also discussed the related literature in knowledge graph learning and multimodal learning in Appendix H.

## 3    METHOD

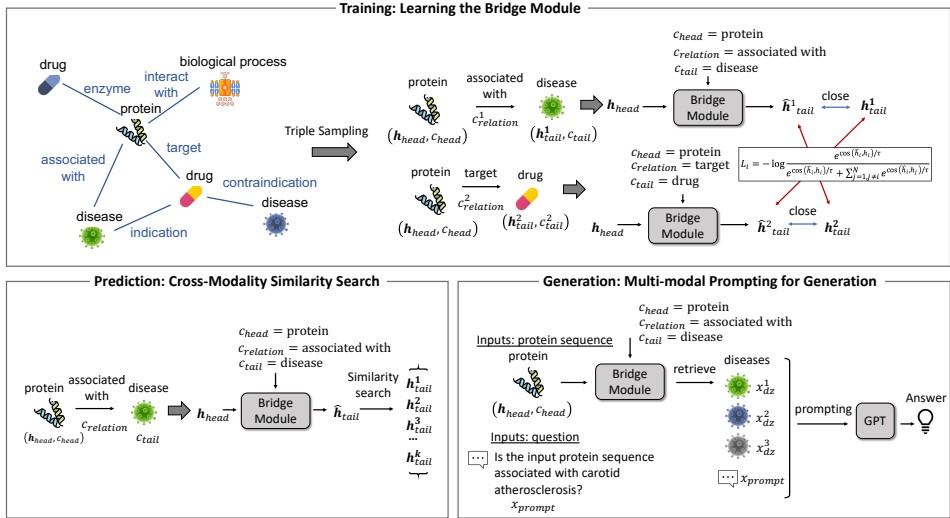

Figure 2: The overall workflow of BioBRIDGE: (1) **top**: we train a bridge module that transforms the head node embedding to the tail node space with contrastive learning. (2) **bottom left**: the trained bridge module enables cross-modal prediction through the similarity search. (3) **bottom right**: The bridge module enables multimodal prompting for retrieval-augmented generation.

The study aims to bridge embeddings across multiple modalities supervised by a KG. The overall workflow of BioBRIDGE is illustrated in Figure 2.

**Knowledge Graph (KG).** KG consists of nodes $\mathcal{V}$ and edges $\mathcal{E}$ as $\mathcal{G} = \{\mathcal{V}, \mathcal{E}\}$. A node in the graph, namely $\mathbf{v}_i = \{\mathbf{x}^v, c^v\}$, where $c^v$ is the node modality and $\mathbf{x}^v$ is the node feature. For example, a protein as a node in the KG is with $c^v$ = "protein" and $\mathbf{x}^v$ = the protein's sequence. An edge $e \in \mathcal{E}$ that connects two nodes $\mathbf{v}_i$ and $\mathbf{v}_j$ is called a *triple* in the context of KG, as $\mathbf{t}_{ij} = \{\mathbf{v}_i, \mathbf{v}_j, r_{ij}\}$, where $r_{ij}$ indicates the relation between the head and tail nodes. KGs present the relations across modalities in a distributed way so BioBRIDGE does not need to set a central modality. Although traditional KG embedding (KGE) methods also enable cross-modal prediction via link prediction, they do not extrapolate to nodes not in the training KG. Instead, BioBRIDGE learns to transform the head-modality FM embeddings to the tail's space. KG is not needed in the testing phase.

**Bridge Modalities**. With foundation models (FMs) pre-trained on unimodal data, we target to bridge the unimodal FMs to fulfill multimodal tasks without fine-tuning the FMs. For two nodes from different modalities, we encode $\mathbf{v}_i$ and $\mathbf{v}_j$ into embeddings: $\mathbf{h}_i = f(\mathbf{x}_i^v)$ and $\mathbf{h}_j = g(\mathbf{x}_j^v)$ where $f(\cdot)$ and $g(\cdot)$ are two FMs hence $\mathbf{h}_i$ and $\mathbf{h}_j$ are in different spaces. We aim to build the transformation that projects $\mathbf{h}_i$ to the space of $\mathbf{h}_j$, as $\hat{\mathbf{h}}_i = \phi(\mathbf{h}_i, c_i^v, c_j^v, r_{ij})$ that considers the modality types of two samples and their relations. As a result, the embedding $\hat{\mathbf{h}}_i$ is aligned with $\mathbf{h}_j$ and can match relevant samples from the modality $c_j^v$ by embedding-based similarity search.

## 3.1 TRAINING & PREDICTION

**Encoding & Transformation**. In the training phase, BioBRIDGE samples a triple $\mathbf{t}_{ij}$ that connects $\mathbf{v}_i$ and $\mathbf{v}_j$ from two modalities $c_i^v$ and $c_j^v$, respectively. The bridge module $\phi$ transforms the head node embedding $\mathbf{h}_i$ to the space of modality $c_j^v$ to yield $\hat{\mathbf{h}}_i$. Specifically, the raw embedding $\mathbf{h}$ of a sample $\mathbf{v} = \{\mathbf{x}, c\}$ is projected by the modality-specific projection head $p$ as $\mathbf{z} = p_c(\mathbf{h}) \in \mathbb{R}^d$ to ensure all embeddings follow the same dimension. We also treat $c_i^v$, $c_j^v$, $r_{ij}$ as categorical variables and generate their embeddings $\mathbf{c}_i$, $\mathbf{c}_j$, and $\mathbf{r}_{ij}$, respectively, which all are in dimension of $d$. The projected head node embedding $\mathbf{z}_i$ is transformed by

$$\hat{\mathbf{h}}_i = \mathbf{z}_i + \psi(\mathbf{z}_i, \mathbf{c}_i, \mathbf{c}_j, \mathbf{r}_{ij}), \tag{1}$$

where $\psi : \mathbb{R}^d \mapsto \mathbb{R}^d$ generates the relation-aware transformation embeddings additive to $\mathbf{z}_i$.

**Loss Function**. We corrupt $\mathbf{t}_{ij}$ by replacing the tail node with others $\{\mathbf{v}_{j'}\}_{j'}$ where $c_{j'}^v = c_j^v$ to build informative negative samples. Based on the encoded tail nodes and the transformed head node, as $\{\hat{\mathbf{h}}_i, \mathbf{z}_j, \underbrace{\mathbf{z}_{j_1}, \dots \mathbf{z}_{j_M}}_{\text{negative tails}}\}$, we perform contrastive learning with InfoNCE loss (Oord et al., 2018) as

$$\mathcal{L}_{ij} = -\log \frac{\exp(\hat{\mathbf{h}}_i \cdot \mathbf{z}_j / \tau)}{\exp(\hat{\mathbf{h}}_i \cdot \mathbf{z}_j / \tau) + \sum_{j' \neq j}^{M} \exp(\hat{\mathbf{h}}_i \cdot \mathbf{z}_{j'} / \tau)}, \tag{2}$$

where $M$ is the number of the sampled negative tails; $\tau$ is a scalar temperature, and all embeddings are normalized by their $\ell_2$-norm before passing to the loss function. This loss pushes $\hat{\mathbf{h}}_i$ close to the positive tail $\mathbf{h}_j$. As the base unimodal FMs are frozen, we only update the parameters of the transformation module $\psi$ and the modality-specific projection head $p_c$.

**Prediction**. Though triples extracted from KGs are used in the training, BioBRIDGE does not refer to KG for its inference. For instance, for $\mathbf{v}_i = \{\mathbf{x}_i^v, c_i^v\}$ and the target modality $\mathcal{C}$, we encode $\mathbf{v}_j, \forall c_j^v \in \mathcal{C}$ by the base FM $g(\cdot)$ and project them into the normalized embeddings $\mathbf{H}_{\mathcal{C}} = \{\mathbf{z}_1, \mathbf{z}_2, \dots, \mathbf{z}_{|\mathcal{C}|}\}$. Then, we encode $\mathbf{v}_i$ with the base FM $f(\cdot)$ and transform it to the embedding space of $\mathcal{C}$, yielding the normalized $\hat{\mathbf{h}}_i$. We can compare the similarity of $\hat{\mathbf{h}}_i$ with $\mathbf{H}_{\mathcal{C}}$ efficiently through matrix inner product, as $\hat{y} = \mathbf{H}_{\mathcal{C}}^{\top} \hat{\mathbf{h}}_i \in [-1, 1]^{|\mathcal{C}|}$.

## 3.2 IMPLEMENTATION

**Dataset**. We draw a subset of PrimeKG (Chandak et al., 2023) to build the training knowledge graph. Specifically, we pick the six main node types from the graph: *Protein*, *Molecule*, *Disease*, *Biological process* (BP), *Molecular function* (MF), and *Cellular component* (CC) without the loss of generality. The statistics of the triples in the training KG are available in Table 8. The exact training set varies depending on the downstream evaluation datasets to avoid data leakage in our experiments. We describe how we curate the training data based on PrimeKG in Appendix B.

**Model**. We categorize the six types of nodes into three modalities: protein sequence, SMILES strings, and natural language. Technically, we utilized ESM2-3B (Lin et al., 2023) to encode proteins, UniMol (Zhou et al., 2023a) to encode drug molecules, and PubMedBERT (Gu et al., 2021) to encode diseases, biological processes, molecular functions, and cellular components. For a text node, we concatenate its name and definition to form the inputs for PubMedBERT.

While there are many potential options to build the transformation, we deliberately choose a vanilla six-layer transformer model for the bridge module $\psi$ in Eq. 1 to verify the plausibility of the method. In detail, we stack $\mathbf{z}_i$, $\mathbf{c}_i$, $\mathbf{c}_j$, and $\mathbf{r}_{ij}$ to build the input $\mathbf{Z} \in \mathbb{R}^{4 \times d}$ for the transformers. We draw the embedding on the first position after the transformer as the output of $\psi$ to add to the input $\mathbf{z}$. Please refer to Appendix I for the hyperparameter setups.

## 3.3 EXISTENCE OF BRIDGE MODULE AND LEARNABILITY

**Assumptions 1** (Assumptions on the neural networks and Knowledge Graph). *1. Let $K \in \mathbb{Z}^+$ be the total number of modalities and let $M_1, \ldots, M_K$ denote the $K$ neural networks trained on the different modalities whose parameters are now frozen.*

2. *For every neural network $M_k : \Omega_k \to S_k, d > 0, k \in \{1, \ldots k\}, S_k \subseteq \mathbb{R}^d, \Omega_i$ denotes the input space/ domain and the output space $S_k$ for every neural network is a linear subspace of $\mathbb{R}^d$. Specifically, we will assume the dimension of every subspace is the same.*
3. *Let $\mathcal{G} = (\mathcal{V}, \mathcal{E})$ denote the knowledge graph where $\mathcal{V}$ is the set of nodes and $\mathcal{E}$ is the set of edges. Each node $v \in \mathcal{V}$ belongs to one of the $K$ modalities.*
4. *Every $e \in \mathcal{E}$ of the knowledge graph which connects two nodes $v_i, v_j \in \mathcal{V}$ has an associated relation type $r_{ij} \in \mathcal{R}$.*

**Theorem 1** (Existence of a Bridge Module). *For any given pair of nodes $v_i, v_j \in \mathcal{V}$ of modality types $k_{v_i}, k_{v_j} \in \{1, \ldots K\}$ and with representations given by their appropriate neural networks $s_{v_i} \in S_{k_{v_i}}, s_{v_j} \in S_{k_{v_j}}$, which are connected by relation type $r_{ij} \in \mathcal{R}$, there exists a bridge module $B : \mathbb{R}^d \times \{1, \ldots, K\} \times \{1, \ldots, K\} \times \mathcal{R} \to \mathbb{R}^d$ such that $B : (s_{v_i}, k_{v_i}, k_{v_j}, r_{ij}) \mapsto s_{v_j}$*

Here, we present a theorem for the existence of a unified bridge module that connects different modalities. Please refer to Appendix A for the proof.

## 4 EXPERIMENT: CROSS-MODALITY PREDICTION

In this section, we perform the experiments to test the *prediction* capabilities of BioBRIDGE. Specifically, the prediction tasks can be categorized into:

- **In-domain entity and relation types**. Both the types of the input entity and the input relation are present in the training knowledge graph, where we conducted two series of experiments: cross-modality retrieval tasks (§4.1) and semantic similarity inference (§4.2).

- **In-domain entity and out-of-domain relation types**. We consider the case where the target relations are absent in the training graph, i.e., out domain. We conducted protein-protein interaction prediction for this case (§4.3).

- **Out-of-domain entity and in-domain relation types**. We also conducted experiments for out-domain entities but in-domain relations: cross-species protein-phenotype matching (§4.4).

### 4.1 CROSS-MODALITY RETRIEVAL TASKS

**Setup**. BioBRIDGE is able to perform cross-modality retrieval by matching the transformed embedding with the candidate samples in the target modality embedding space. To gauge the quality of the transformed embeddings, we compare our methods with a suite of knowledge graph embedding (KGE) methods: TransE (Bordes et al., 2013), TransD (Ji et al., 2015), TransH (Wang et al., 2014), TransR (Lin et al., 2015), ComplEx (Trouillon et al., 2016), DistMult (Yang et al., 2015), and RotatE (Sun et al., 2019), implemented with OpenKE (Han et al., 2018).

**Metric**. We refer to the KG link prediction literature to use Hit@$K$ ($K \in \{1, 3, 10\}$), and Mean reciprocal rank (MRR) to evaluate the prediction performance. MRR is the average reciprocal rank of all the positive test triples among the corrupted negative triples. Hit@$K$ measures the proportion of positive tail entities among the ranked top-$K$ possible candidates. We calculate these metrics on the direction of tail entity prediction.

**Dataset & Baseline**. We split the raw PrimeKG triples to set up the cross-modality retrieval tasks. For each type of triple, we randomly sample 80%, 10%, and 10% for the train, validation, and test sets, respectively. Then, we separate the test set by triple types, with a special focus on the

predictions for: {*Protein, BP/MF/CC, Interacts with*}, {*Drug, Disease, Indication*}, {*Drug, Protein, Target*}, {*Protein, Disease, Associated with*}, {*Drug, Disease, Contraindication*}. The statistics of the train/valid/test data used in this experiment are available in Table 10.

Table 1: Mean reciprocal rank (MRR) on the seven cross-modal prediction tasks. "Drug ↛ Disease" indicates the "contraindication" relation between drug and disease. The best are in bold.

| Method | Protein → BP | Protein → MF | Protein → CC | Drug → Disease | Protein → Drug | Disease → Protein | Drug ↛ Disease |
|---|---|---|---|---|---|---|---|
| TransE | 0.034 | 0.046 | 0.044 | 0.017 | 0.033 | 0.024 | 0.010 |
| TransR | 0.045 | 0.060 | 0.048 | 0.053 | 0.069 | 0.028 | 0.029 |
| TransH | 0.044 | 0.061 | 0.057 | 0.026 | 0.043 | 0.024 | 0.014 |
| TransD | 0.043 | 0.059 | 0.053 | 0.022 | 0.049 | 0.024 | 0.013 |
| ComplEx | 0.084 | 0.100 | 0.099 | 0.042 | 0.079 | 0.059 | 0.048 |
| DistMult | 0.054 | 0.089 | 0.095 | 0.025 | 0.044 | 0.033 | 0.047 |
| RotatE | 0.079 | 0.119 | 0.107 | 0.150 | 0.125 | 0.070 | 0.076 |
| BioBRIDGE | **0.136** | **0.326** | **0.319** | **0.189** | **0.172** | **0.084** | **0.081** |

**Result**. We show MRR across the seven tasks in Table 1 and the breakdown performances in Tables 11, 12, 13, 14, 15, 16, and 17, respectively. We further report the overall average ranking of all methods across these tasks in Table 9. We found that BioBRIDGE is consistently ranked the best among the KGE methods. The specialized KGE algorithms learn the node and relation embeddings from scratch exclusively based on the KG, while our method builds on pre-trained FMs that already possess rich prior knowledge. As such, BioBRIDGE bridges modalities in a much more data-efficient way. Breaking down into the performance on different tasks, as shown in Table 1, we observed that BioBRIDGE gains a higher margin over baselines on tasks with fewer triples from the KG. For instance, BioBRIDGE is around 3× better than the best baseline for "Protein → MF" while is around 1.6× better for "Protein → BP", which signals the benefit of BioBRIDGE in bridging FMs with limited data for multimodal tasks over training a multimodal model from scratch.

## 4.2 SEMANTIC SIMILARITY INFERENCE

**Setup & Metric**. The objective of this analysis is to evaluate the extent to which the encoded protein embeddings can capture biomolecular functional similarity, i.e., biological process (BP), molecular function (MF), and cellular component (CC). We follow the experimental protocol in (Unsal et al., 2022) that takes the gene ontology (GO) terms annotations of proteins as the target. For our method, we use the protein embeddings transformed to the BP, MF, and CC spaces as the input for evaluation. We compute the pairwise Manhattan Similarities of the encoded protein embeddings as the predictions. The final score is obtained by computing the Spearman's rank correlation between the predictions and the flattened groundtruth matrix, which is the larger, the better.

**Dataset & Baseline**. We leverage the test sets released by (Zhou et al., 2023b) where three $500 \times 500$ labeled matrices store the pairwise Lin Similarities of the protein associated BP, MF, and CC, respectively. We aggregate these matrices to obtain 1,123 unique protein sequences and remove them from the training knowledge graph to avoid data leakage. We compare our method with the following baselines: MSA Transformer (Rao et al., 2021), ESM-1B (Rives et al., 2021), ProtT5-XL (El-naggar et al., 2021), ESM2-3B (Lin et al., 2023), OntoProtein (Zhang et al., 2022), and KeAP (Zhou et al., 2023b).

Table 2: The comparison on semantic similarity inference across methods. The best are in bold. "Avg" is short for the average of results.

| Method | MF | BP | CC | Avg |
|---|---|---|---|---|
| MSA Transformer | 0.38 | 0.31 | 0.30 | 0.33 |
| ProtT5-XL | 0.57 | 0.21 | 0.40 | 0.39 |
| ProtBERT | 0.41 | 0.35 | 0.36 | 0.37 |
| ESM-1B | 0.38 | 0.42 | 0.37 | 0.39 |
| ESM2-3B | 0.33 | 0.42 | 0.23 | 0.32 |
| OntoProtein | 0.41 | 0.36 | 0.36 | 0.38 |
| KeAP | 0.41 | 0.41 | 0.40 | 0.41 |
| BioBRIDGE | **0.91** | **0.80** | **0.73** | **0.81** |

**Result**. We report the results in Table 2, where our method yields a substantial improvement, with around 2× better than the best baseline on average. Across the baselines, we observed the methods augmented by KG, including KeAP and OntoProtein, yield better results than the others, implying that KG connecting proteins and the biological attributes enhance protein representation learning. Nonetheless, BioBRIDGE learns to transform the protein embeddings to the biomolecular functional embedding space, thus aligning protein sequences better with the semantic meaning of func-

tional terms. Also, the involvement of other modalities, like drugs from the KG in training, further enriches the supervision for the transformation model.

## 4.3 PROTEIN-PROTEIN INTERACTION

Table 3: The F1 scores of the selected methods on the protein-protein interaction task with three datasets. "B+D" is short for the mean of BFS and DFS results. The best results are in bold.

| Method | Split ID | SHS27K | | | SHS148K | | | STRING | | |
|---|---|---|---|---|---|---|---|---|---|---|
| | | BFS | DFS | B+D | BFS | DFS | B+D | BFS | DFS | B+D |
| KeAP | 0 | 0.695 | 0.744 | | 0.634 | 0.831 | | 0.801 | 0.884 | |
| | 1 | 0.764 | 0.713 | | 0.626 | 0.813 | | 0.783 | 0.886 | |
| | 2 | 0.694 | 0.790 | | 0.629 | 0.825 | | 0.765 | 0.884 | |
| | Avg (Std) | 0.718 (0.040) | 0.749 (0.039) | 0.733 | 0.630 (0.004) | 0.823 (0.009) | 0.726 | 0.783 (0.018) | 0.885 (0.001) | 0.834 |
| ESM2-3B | 0 | 0.687 | 0.733 | | 0.637 | 0.839 | | 0.801 | 0.880 | |
| | 1 | 0.768 | 0.725 | | 0.644 | 0.827 | | 0.781 | 0.884 | |
| | 2 | 0.690 | 0.791 | | 0.632 | 0.819 | | 0.778 | 0.881 | |
| | Avg (Std) | 0.715 (0.046) | 0.750 (0.036) | 0.732 | 0.638 (0.006) | 0.828 (0.010) | 0.733 | **0.787 (0.013)** | 0.882 (0.002) | 0.834 |
| BioBRIDGE | 0 | 0.704 | 0.764 | | 0.645 | 0.840 | | 0.791 | 0.890 | |
| | 1 | 0.765 | 0.722 | | 0.638 | 0.834 | | 0.787 | 0.891 | |
| | 2 | 0.701 | 0.778 | | 0.647 | 0.829 | | 0.769 | 0.888 | |
| | Avg (Std) | **0.723 (0.036)** | **0.755 (0.029)** | **0.739** | **0.643 (0.005)** | **0.834 (0.005)** | **0.739** | 0.782 (0.012) | **0.890 (0.002)** | **0.836** |

**Setup & Metric**. We study the protein-protein interaction (PPI) prediction task because it represents the second experiment setup: in-domain entity and out-of-domain relation. The PPI prediction task aims to classify 7 interaction types of a pair of proteins: *reaction*, *binding*, *post-translational modifications (ptmod)*, *activation*, *inhibition*, *catalysis*, and *expression*. Although the `ppi` relation is present in PrimeKG, it only represents the physical interaction (similar to "Binding" in the seven types), while the other six types are out-of-domain.

Following the setup of (Zhang et al., 2022), we extract the protein embeddings with the baseline pre-trained protein models, which serve as the input for a graph neural network model to be trained on the PPI network. Our method uses the protein embeddings transformed to protein space with the relation `ppi`. We report the F1 score for this multi-class classification task.

**Dataset & Baseline**. Two baselines are selected for comparison: ESM2-3B (Lin et al., 2023) and KeAP (Zhou et al., 2023b). We test them on three PPI datasets: SHS27K (Chen et al., 2019), SHS148K (Chen et al., 2019), and STRING (Lv et al., 2021). Following the setup in (Zhou et al., 2023b), we perform Breadth-First Search (BFS) and Depth-First Search (DFS) to generate two train/validation/test splits, respectively.

**Result**. From the results in Table 3, we observe that though the results vary across splits, our method shows a consistent improvement over the baselines in most scenarios. It is illustrated that ESM2-3B performs better than the prior state-of-the-art KeAP, which can be attributed to its pre-training on an enormous protein database. `BioBRIDGE` further enhances the embeddings of ESM2 by injecting the relation "ppi", and then transforms back to the protein space. `BioBRIDGE` exhibits greater benefit on the datasets with fewer samples like SHS27K as it enriches the protein embedding with the protein-protein interaction ontology information. When the number of training data increases, all methods tend to converge to the same level while the baselines are still inferior to `BioBRIDGE`.

## 4.4 CROSS-SPECIES PROTEIN-PHENOTYPE MATCHING

**Setup & Metric**. We propose this novel task to test the capability of `BioBRIDGE` to handle the cross-modality transformation for out-of-domain entity and in-domain relation. As PrimeKG only contains human proteins, we build a dataset of mouse proteins and the associated mouse phenotypes from the Mouse Genome Informatics (MGI) resource (Eppig et al., 2017), acting as out-of-domain entities. We elaborate on the curation process of this data in Appendix E. Since the modality "phenotype" is absent in `BioBRIDGE`'s training data, we transform the encoded mouse protein embeddings to the "disease" space with the relation "associate with". We use a suite of ranking metrics to evaluate the matching performance, including Recall@K, Precision@K, and nDCG@K.

**Dataset**. We build two datasets for 1) matching mouse protein (MG) to mouse phenotype (MP) and 2) matching mouse protein (MG) to human phenotype (HP). The data statistics are available in Table 18. Basically, there are 28 mouse phenotypes to predict for Task 1 and 353 human phenotypes for Task 2, respectively.

**Baseline**. There is no previous work for this task. We create a dual-encoder model with a protein encoder: ESM2-1B (Lin et al., 2023) and a text encoder: PubMedBERT (Gu et al., 2021). To build the supervised baseline, we perform contrastive learning with the dual-encoder model on the paired mouse protein and phenotype.

Table 4: The comparison between the zero-shot BioBRIDGE ("0-shot") and the supervised baseline on the cross-species retrieval tasks. "Diff" indicates the relative improvement of BioBRIDGE over the supervised baseline.

| Task 1: Mouse Protein to Mouse Phenotype | | | | Task 2: Mouse Protein to Human Phenotype | | | |
|---|---|---|---|---|---|---|---|
| Metric | Supervised | BioBRIDGE-0-shot | Diff | Metric | Supervised | BioBRIDGE-0-shot | Diff |
| Rec@1 | 0.061 | **0.067** | 10% | Prec@1 | 0.001 | **0.043** | 3351% |
| Prec@1 | **0.360** | 0.338 | -6% | Rec@1 | 0.001 | **0.022** | 3098% |
| Rec@3 | 0.145 | **0.185** | 28% | Prec@5 | 0.003 | **0.022** | 624% |
| Prec@3 | 0.289 | **0.316** | 9% | Rec@5 | 0.011 | **0.053** | 393% |
| Rec@5 | 0.194 | **0.278** | 43% | Rec@10 | 0.026 | **0.088** | 236% |
| Prec@5 | 0.233 | **0.292** | 25% | Rec@20 | 0.051 | **0.168** | 228% |
| nDCG@5 | 0.293 | **0.350** | 20% | nDCG@20 | 0.019 | **0.082** | 330% |

**Result**. We report the results in Table 4. This is a challenging task because neither mouse proteins nor mouse/human phenotypes were used in its training data. Despite this, the underlying protein FM underwent comprehensive pre-training on protein sequences across various species. As BioBRIDGE learns to bridge human protein and human diseases, it demonstrates the emergent ability to transform mouse protein from the protein space to mouse phenotype in the text space.

In Task 1, BioBRIDGE showcases a large margin over the supervised baseline that was fine-tuned on the paired mouse protein and phenotype data. This observation underscores the feasibility of transferring BioBRIDGE to a novel domain without further training. In Task 2, the supervised baseline failed to extrapolate to match mouse protein to human phenotype despite being learned on paired mouse proteins and mouse phenotypes. However, BioBRIDGE leverages the prior knowledge of the base FMs on protein and human disease. This inspiring result hints at the potential for novel bioinformatic analysis based on the bridged FMs enabled via cross-modal matching.

# 5 CASE STUDY: MULTIMODAL GENERATION

Table 5: Case study of multi-modal Q&A based on BioBRIDGE: Input a molecule/protein and a question, retrieve the relevant diseases and proteins/gene ontology terms, and prompt LLM to answer. The references are drawn for the input molecule/protein-based drug pages from DrugBank.

| Input Molecule | Question | Retrieved Disease | Retrieved Protein | Answer | Reference |
|---|---|---|---|---|---|
| CC[C@@]1(O)C[C@@H]2CN(C1) CCC1=C(NC3=CC=CC=C13)[C@@](C2) (C(=O)OC)C1=C(OC)C=C2N(C=O)[C@@H]3 [C@]4(CCN5CC=C[C@](CC)([C@@H]45) [C@@H](OC(C)=O)[C@]3(O)C(=O)OC)C2=C1 | What are the associated conditions and pharmacodynamics of this drug? | yolk sac tumor
mediastinal malignant lymphoma
non-small cell lung carcinoma (disease)
Burkitt lymphoma
malignant sex cord stromal tumor of ovary
B-cell neoplasm
dysgerminoma of ovary
sarcoma
duodenum cancer
malignant Sertoli-Leydig cell tumor of ovary | TUBB8
TUBB
TUBA4A
TUBB4B
TUBB2B
TUBB4A
TUBB3
TUBB2A
TUBA1C
TUBA3C | [...] the main associated conditions of this small molecule drug could be cancers of the yolk sac , [...], malignant Sertoli-Leydig cell tumor of ovary . The main pharmacodynamics of this small molecule drug could be its ability to bind and inhibit the activity of some of the tubulin proteins , such as TUBB8, [...] and TUBA3C. By inhibiting these proteins, the drug could disrupt the normal function of the microtubules , which are filament-like structures that support the cell structure and participate in cell division. This could lead to cell death and make the cancer cells more susceptible to other treatments. | Vincristine is a vinca alkaloid used to treat acute leukemia , malignant lymphoma , Hodgkin's disease , acute erythraemia , and acute panmyelosis . The anti-tumor activity of Vincristine is thought to be due primarily to inhibition of mitosis at metaphase through its interaction with tubulin . Ref: DrugBank: DB00541 |

| Input Protein | Question | Retrieved Disease | Retrieved GO Terms | Answer | Reference |
|---|---|---|---|---|---|
| MPTSSSTKKTQLQLEHLLLDLQMILNGI NNYKNPKLTRMLTFKFYMPKKATELKHL QCLEEELKPLEEVLNLAQSKNFHLRPRDL ISNINVIVLELKGSETTFMCEY ADETATIVEFLNRWITFS QSIISTLT | Describe the pharmacology of this protein-based therapy. | hepatocellular carcinoma
hepatocellular adenoma
adult hepatocellular carcinoma
hepatocellular clear cell carcinoma
liver cancer
undifferentiated carcinoma of liver and intrahepatic biliary tract
nonpapillary renal cell carcinoma
squamous cell carcinoma of liver and intrahepatic biliary tract
familial prostate carcinoma
prostate cancer
adenocarcinoma of liver and intrahepatic biliary tract
drug-induced liver injury | positive regulation of cell population proliferation
multicellular organism development
signal transduction
spermatogenesis
cell differentiation
G protein-coupled receptor signaling pathway
negative regulation of cell population proliferation | [...] some possible associated conditions of this protein-based therapy are hepatocellular carcinoma , [...] and intrahepatic biliary tract . The main pharmacodynamics of this therapy could be related to the regulation of cell population proliferation , cell differentiation, and signal transduction, which are all important processes involved in the growth and development of cancer cells. The protein sequence may also have some interactions with metal ions, proteins, and proteases, which could impact its ability to bind to and modulate the activity of other proteins involved in cancer development. | Used to treat renal cell carcinoma . Aldesleukin induces the enhancement of lymphocyte mitogenesis and stimulation of long-term growth of human interleukin-2 dependent cell lines. Ref: DrugBank:DB00041 |

In this section, we show how BioBRIDGE supports a multimodal generation system through cross-modal retrieval. We leverage Galactica-30B (Taylor et al., 2022) that was fine-tuned on instruction datasets as the base generator[2]. We prompt Galactica to answer the input question or generate target molecules following the input instruction. The used prompts are in Appendix G.

---

[2]Huggingface: GeorgiaTechResearchInstitute/galactica-30b-evol-instruct-70k

**Multiodal Question & Answer**. `BioBRIDGE` accommodates multimodal input that consists of a molecule SMILES string/protein sequence with an associated question in natural language. In this task, `BioBRIDGE` serves as a cross-modal retriever to enhance the contexts for Galactica's response to the input question. Particularly, it retrieves the following: (1) identify the potential protein targets of the input molecule, (2) identify the disease indications for the input molecule, (3) identify associated diseases for the input protein, and (4) identify gene ontology terms related to the protein. We choose the drugs from DrugBank that are not in the `BioBRIDGE`'s training KG. We also involve *investigational* drugs to test if `BioBRIDGE` can aid in proposing mechanisms-of-action.

Results are shown in Table 5, with more results in Table 19 in the appendix. It demonstrates that `BioBRIDGE` provides key evidence that prompts Galactica to reach the right answer. For instance, `BioBRIDGE` can pinpoint a group of tubulin proteins and oncology-related conditions for Vincristine. This process enables Galactica to provide an accurate response, indicating that this drug inhibits the mitosis of cancer cells. In addition, in Table 19, the investigational drug Rimacalib, which has been used in trials studying the treatment of rheumatoid arthritis, is identified by our method to possess immunomodulatory, anti-inflammatory, and anti-arthritic effects. It hence prompts Galactica to reach the answer that this drug may treat diseases such as rheumatoid arthritis.

Table 6: Case study of multi-modal generation based on `BioBRIDGE`: Input the target condition and the intended mechanism, retrieve the relevant proteins, and prompt LLM to generate the target small molecule drug.

| Target Condition | Target Effect | Retrieved Protein | Generated Molecule | Most Similar Drug |
|---|---|---|---|---|
| malignant lymphoma | The inhibition of mitosis at metaphase of cancer cells via polychemotherapy. | MTHFR BCL2 TYMS CAT CASP8 CSF3 CDKN2A TP53 CREBBP SOD2 |  C#CC(C)(C)NC(=O)C1=CC(Cl)=CC(Cl)=C1 |  [DB01168] Procarbazine CNNCC1=CC=C(C=C1)C(=O)NC(C)C Procarbazine is an antineoplastic agent indicated for the treatment of stage III and stage IV Hodgkin's disease in combination with other chemotherapeutic agents. |

**Multi-Modal Generation**. This task aims to achieve the text-guided generation of small-molecule drugs. We use `BioBRIDGE` to enrich the context for Galactica by retrieving the target proteins that are possibly associated with the target conditions. To validate the structural integrity of the generated molecules, we utilize RDKit to calculate the Tanimoto distance between the generated SMILES string and all candidate small molecule drugs listed in DrugBank. We then identify the most similar drugs. Results are shown in Table 6 with more in Table 20. We also made a baseline that prompts Galactica to generate the molecule directly, shown in Table 21. We found that (1) `BioBRIDGE` prompts Galactica to generate valid drug molecules; (2) the generated molecule usually shares similarities with the real drugs that were considered effective for the target condition; (3) Prompting Galactica without RAG leads to poor generation results. For instance, in Table 6, the model-generated drug is most similar to Procarbazine, which is used to treat stage III/IV Hodgkin's disease in chemotherapy by impeding the division of cancer cells. This implies the generated drug probably fits the proposed target effect and treats lymphoma.

## 6 CONCLUSION

This paper investigated bridging unimodal biomedical foundation models (FM) for multimodal tasks. We identified that `BioBRIDGE` can effectively transform the embeddings to the target modality, considering the types of source modality, target modality, and their relations. It is with great parameter efficiency: only the bridge module needs training while all the base FMs are kept fixed, supervised by the relational information from biomedical knowledge graphs. We also identified that `BioBRIDGE` can handle a diverse set of cross-modal prediction tasks by extrapolating to in-domain/out-of-domain entities and relations. And the yielded performances are on par with the supervised specialist models in each task. In addition, we demonstrated how the bridged FMs can support generation tasks with multimodal inputs. In the future, we envision that `BioBRIDGE` can be extended to connect pre-trained FMs from other domains as long as entities across different modalities can be represented in a KG.

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

## A  PROVING EXISTENCE OF BRIDGE MODULE AND LEARNABILITY

This section aims to show the existence of a bridge module, followed by its learnability. Given our setup of unimodal FMs, we will show the existence of transformation modules that can take us from any modality's output space to another under certain assumptions. We will proceed to first state our assumptions and then the statement, followed by the proof for the statement.

**Assumptions 1** (Assumptions on the neural networks and Knowledge Graph). *1. Let $K \in \mathbb{Z}^+$ be the total number of modalities and let $M_1, \ldots, M_K$ denote the $K$ neural networks trained on the different modalities whose parameters are now frozen.*

*2. For every neural network $M_k : \Omega_k \to S_k, d > 0, k \in \{1, \ldots k\}, S_k \subseteq \mathbb{R}^d, \Omega_i$ denotes the input space/ domain and the output space $S_k$ for every neural network is a linear subspace of $\mathbb{R}^d$. Specifically, we will assume the dimension of every subspace is the same.*

*3. Let $\mathcal{G} = (\mathcal{V}, \mathcal{E})$ denote the knowledge graph where $\mathcal{V}$ is the set of nodes and $\mathcal{E}$ is the set of edges. Each node $v \in \mathcal{V}$ belongs to one of the $K$ modalities.*

*4. Every $e \in \mathcal{E}$ of the knowledge graph which connects two nodes $v_i, v_j \in \mathcal{V}$ has an associated relation type $r_{ij} \in \mathcal{R}$.*

**Theorem 1** (Existence of a Bridge Module). *For any given pair of nodes $v_i, v_j \in \mathcal{V}$ of modality types $k_{v_i}, k_{v_j} \in \{1, \ldots K\}$ and with representations given by their appropriate neural networks $s_{v_i} \in S_{k_{v_i}}, s_{v_j} \in S_{k_{v_j}}$, which are connected by relation type $r_{ij} \in \mathcal{R}$, there exists a bridge module $B : \mathbb{R}^d \times \{1, \ldots, K\} \times \{1, \ldots, K\} \times \mathcal{R} \to \mathbb{R}^d$ such that $B : (s_{v_i}, k_{v_i}, k_{v_j}, r_{ij}) \mapsto s_{v_j}$*

*Proof.* At a very high level, a direct consequence of the Hahn-Banach Theorem (Conway, 2019) can be used to see that given a Euclidean space and given two compact, convex subsets of that Euclidean space, there exists a continuous function mapping between them (given that there is a linear functional that separates them). Given our assumption that our subspaces are linear and therefore convex, as well as the assumption of continuity of neural networks, we can quickly assert the existence of transformations between the spaces.

Here, however, we will go into the details and show a simple proof using linear algebra. We will use a multi-step approach to prove the existence of the bridge module. Firstly, we will look at expressibility in terms of linear subspaces and then introduce conditional transformations based on relation types. Subsequently, we will look at the compositionality and aggregation of paths (as we are dealing with knowledge graphs) for the bridge module. We will specifically look at linear subspaces here, but similar arguments can also be established for convex hull-like spaces or other spaces that have an established geometry.

First, we look at the linear mapping between linear subspaces. Given any two linear subspaces $S_x, S_y$, a linear transformation between the two is one which maps points in $S_x$ to $S_y$ i.e. $T_{xy} : S_x \to S_y$ - and this can always be represented using a matrix $A_{xy}$ as $T_{xy}(p) = A_{xy} \cdot p, p \in S_x$. The existence of such a transformation is guaranteed by the rank nullity theorem (Strang, 2012) without any loss of information (as the dimensions of the subspaces are taken to be the same by construction).

Next, any relation in $r_{ij} \in \mathcal{R}$ can be represented as a vector, i.e., a relation embedding $r(r_{ij}) \in \mathbb{R}^p, p > 0$ and the transformation $T$ between subspaces can be a function over the matrix $A_{xy}$ as well as $r(r_{ij})$ i.e. $A_{xy}^a = f(A_{xy}, r(r_{ij}))$ and the conditional transformation (relation dependent) of any point $p \in S_x$ via relation $r_{ij}$ is now given as $T_{xy}^{r_{ij}}(p) = A_{xy}^{r_{ij}} \cdot p$. Such a formulation also allows us to directly model compositionality of transformations (due to the properties of matrices) as given two transformations $T_{xy}, T_{yz}$ between $S_x \to S_y \to S_z$, a composition $T_{yz} \circ T_{xy}$ (given by $A_{yz}^{r_{jk}} \cdot A_{xy}^{r_{ij}}$ in matrix notation) captures a transformation from $S_x$ to $S_z$. Given that we are dealing with linear subspaces and transformations, these compositions are always defined (Strang, 2012).

Next, we look at the aggregation of multiple paths - since we are dealing with KGs, there can be multiple paths between any two nodes. In such cases, the transformation between any two subspaces can then be obtained as a weighted combination of the paths, i.e., $A_{agg} = \Sigma_{i=1}^N w_i A_i$ such that $\Sigma_{i=1}^N w_i = 1$ where $w_i$ is the weight associated with a path and $A$ gives the transformation matrix between the spaces. When all $w_i > 0$, this forms a convex combination and ensures that the aggregated transformation is still within the space by the original transformations (Boyd & Vandenberghe, 2004) and therefore confirms the existence of a unified bridge module.  □

Now, given the existence of the above transformations as defined by the KG, the convergence of sequences of linear transformations (Halmos, 2017), the universality of neural networks (Hornik, 1991; Cybenko, 1989) and the continuity of the unimodal FM's ensure that such conditional transformations (given by relation types) can be approximated when appropriate loss functions are employed via SGD.

## B ESTABLISHING TRAINING DATA BASED ON PRIMEKG

Table 7: The statistics of nodes and edges of the raw PrimeKG data and the processed KG for training `BioBRIDGE`. "Original": the original PrimeKG; "Processed": the processed KG; "Dropped": how many samples are dropped in the processing.

| Nodes | | | | |
| --- | --- | --- | --- | --- |
| Modality | Original | Processed | Dropped | Percent dropped |
| biological process | 28,642 | 27,478 | 1,164 | 4.06% |
| protein | 27,671 | 19,162 | 8,509 | 30.75% |
| disease | 17,080 | 17,080 | 0 | 0.00% |
| molecular function | 11,169 | 10,966 | 203 | 1.82% |
| drug | 7,957 | 6,948 | 1,009 | 12.68% |
| cellular component | 4,176 | 4,013 | 163 | 3.90% |
| Summation | 96,695 | 85,647 | 11,048 | 11.43% |
| **Edges** | | | | |
| Relation type | Original | Processed | Dropped | Percent Dropped |
| drug_drug | 2,672,628 | 2,241,466 | 431,162 | 16.13% |
| protein_protein | 642,150 | 629,208 | 12,942 | 2.02% |
| bioprocess_protein | 289,610 | 272,642 | 16,968 | 5.86% |
| cellcomp_protein | 166,804 | 149,504 | 17,300 | 10.37% |
| disease_protein | 160,822 | 155,924 | 4,898 | 3.05% |
| molfunc_protein | 139,060 | 133,522 | 5,538 | 3.98% |
| bioprocess_bioprocess | 105,772 | 99,630 | 6,142 | 5.81% |
| disease_disease | 64,388 | 64,388 | 0 | 0.00% |
| contraindication | 61,350 | 60,130 | 1,220 | 1.99% |
| drug_protein | 51,306 | 47,614 | 3,692 | 7.20% |
| molfunc_molfunc | 27,148 | 26,436 | 712 | 2.62% |
| indication | 18,776 | 17,578 | 1,198 | 6.38% |
| cellcomp_cellcomp | 9,690 | 9,200 | 490 | 5.06% |
| off-label use | 5,136 | 4,998 | 138 | 2.69% |
| Summation | 4,414,640 | 3,912,240 | 502,400 | 11.38% |

This paper aims to make a proof of concept in bridging uni-modal FMs, so it simplifies the KG in the experiment by only keeping six modalities. Nonetheless, protein, disease, drug, and gene ontology terms are the main biomedical entities covering the majority of real-world biomedical tasks (e.g., drug discovery, repurposing, PPI, protein function prediction, drug target identification, drug-target interaction prediction).

The raw PrimeKG provides a bind of biomedical entities from a widespread of sources. However, it does not provide the associated properties for all the entities, e.g., the sequence structure of the proteins. To train `BioBRIDGE`, we try to link the entities to the external knowledge bases and filter out those without the required property. The statistics of raw KG and the processed are available in Table 7. We describe the data curation process for each type of entities as the following.

**Protein Data**. There are 27,671 proteins in total in the original PrimeKG. We try to match the provided NCBI gene ID of the proteins to the UniProtKB/Swiss-Prot sequence database, via the uniprot id-mapping tool https://www.uniprot.org/id-mapping. We were able to retrieve 27,478 protein sequences that are matched with the gene ID.

We delve deeper into the unmapped proteins and find most are non-protein-coding genes (pseudogenes, rRNA, ncRNA genes), which are genes unable to produce proteins. It is hence reasonable to ignore them for protein-centric tasks.

**Drug Data**. There are 7,957 drugs in the original PrimeKG. We try to match the offered DrugBank ID to the database https://go.drugbank.com/drugs. We dropped drugs without SMILES strings to obtain a total of 6,948 drugs.

**Gene Ontology Terms**. Biological process, molecular function, and cellular component are represented by all gene ontology (GO) terms. We leveraged AmiGO for searching the detailed descriptions of the GO terms through their IDs https://amigo.geneontology.org/amigo/search/ontology. We were able to retrieve the features of the most GO terms then kept 27,478 BP, 10,966 MF, and 4,013 MF, in the training data.

**Disease**. The descriptions of those diseases are provided by the raw PrimeKG data so we are able to keep all 17,080 diseases for training.

## C  DATA STATISTICS OF THE KNOWLEDGE GRAPH

Table 8: The data statistics of the training knowledge graph drawn from PrimeKG.

| Head Node | Tail Node | Relation | Number |
|---|---|---|---|
| protein | protein | ppi | 642,150 |
| | biological process | interacts with | 144,805 |
| | cellular component | interacts with | 83,402 |
| | molecular function | interacts with | 69,530 |
| | disease | associated with | 80,411 |
| | drug | target | 16,380 |
| | | enzyme | 5,317 |
| | | transporter | 3,092 |
| | | carrier | 864 |
| drug | drug | synergistic interaction | 2,672,628 |
| | disease | contraindication | 30,675 |
| | | indication | 9,388 |
| | | off-label use | 2,568 |
| | protein | target | 16,380 |
| | | enzyme | 5,317 |
| | | transporter | 3,092 |
| | | carrier | 864 |
| disease | disease | parent-child | 64,388 |
| | gene/protein | associated with | 80,411 |
| | drug | contraindication | 30,675 |
| | | indication | 9,388 |
| | | off-label use | 2,568 |

## D  CROSS-MODALITY RETRIEVAL TASKS

Table 9: The mean and standard deviation (in parenthesis) of the ranks of all compared methods across the seven cross-modality retrieval tasks. Lower rank is better. Best are in bold.

| Method | Rank-MRR ($\downarrow$) | Rank-Hits@10 ($\downarrow$) | Rank-Hits@3 ($\downarrow$) | Rank-Hits@1 ($\downarrow$) |
|---|---|---|---|---|
| TransE | 7.71 (0.00) | 7.71 (0.00) | 7.43 (0.52) | 5.00 (0.00) |
| TransR | 5.00 (1.41) | 4.57 (1.63) | 5.14 (1.47) | 5.00 (0.00) |
| TransH | 5.86 (0.82) | 5.71 (0.55) | 6.00 (0.41) | 5.00 (0.00) |
| TransD | 6.71 (0.84) | 6.57 (0.82) | 6.57 (0.82) | 5.00 (0.00) |
| ComplEx | 3.00 (0.63) | 3.43 (0.55) | 3.00 (0.63) | 2.86 (0.63) |
| DistMult | 4.57 (1.03) | 5.00 (1.67) | 4.00 (1.79) | 3.14 (0.89) |
| RotatE | 2.14 (0.41) | 2.00 (0.00) | 2.43 (0.84) | 2.86 (0.98) |
| BioBRIDGE | **1.00 (0.00)** | **1.00 (0.00)** | **1.00 (0.00)** | **1.00 (0.00)** |

Table 10: The data statistics of the train/valid/test splits used in cross-modality retrieval tasks.

| Triple Type | # Head Node | # Tail Node | # Train Triple | # Valid Triple | # Test Triple |
|---|---|---|---|---|---|
| Protein - Interacts with - BP | 16,808 | 12,047 | 111,649 | 11,620 | 13,052 |
| Protein - Interacts with - MF | 17,289 | 4,284 | 55,252 | 5,438 | 6,071 |
| Protein - Interacts with - CC | 17,211 | 1,684 | 61,471 | 6,250 | 7,031 |
| Protein - Associated with - Disease | 8,548 | 5,566 | 63,869 | 6,659 | 7,434 |
| Drug - Indication - Disease | 1,592 | 1,279 | 7,290 | 704 | 795 |
| Drug - Contraindiation - Disease | 1,217 | 1,181 | 24,423 | 2,676 | 2,966 |
| Drug - Target - Protein | 5,074 | 2,620 | 12,952 | 920 | 1,055 |

Table 11: Cross-modality prediction performance for triple type: "Protein - Interacts with - BP".

| Method | MRR | Hit@10 | Hit@3 | Hit@1 | MRR Rank | Hit@10 Rank | Hit@3 Rank | Hit@1 Rank |
|---|---|---|---|---|---|---|---|---|
| TransE | 0.034 | 0.097 | 0.020 | 0.000 | 8 | 8 | 8 | 5 |
| TransR | 0.045 | 0.136 | 0.023 | 0.000 | 5 | 4 | 5 | 5 |
| TransH | 0.044 | 0.129 | 0.023 | 0.000 | 6 | 6 | 6 | 5 |
| TransD | 0.043 | 0.125 | 0.020 | 0.000 | 7 | 7 | 7 | 5 |
| ComplEx | 0.084 | 0.218 | 0.067 | 0.020 | 2 | 3 | 2 | 2 |
| DistMult | 0.054 | 0.132 | 0.045 | 0.013 | 4 | 5 | 4 | 3 |
| RotatE | 0.079 | 0.239 | 0.057 | 0.012 | 3 | 2 | 3 | 4 |
| BioBRIDGE | 0.136 | 0.282 | 0.146 | 0.066 | 1 | 1 | 1 | 1 |

Table 12: Cross-modality prediction performance for triple type: "Protein - Interacts with - MF".

| Method | MRR | Hit@10 | Hit@3 | Hit@1 | MRR Rank | Hit@10 Rank | Hit@3 Rank | Hit@1 Rank |
|---|---|---|---|---|---|---|---|---|
| TransE | 0.046 | 0.143 | 0.030 | 0.000 | 8 | 8 | 8 | 5 |
| TransR | 0.060 | 0.196 | 0.041 | 0.000 | 6 | 6 | 5 | 5 |
| TransH | 0.061 | 0.199 | 0.040 | 0.000 | 5 | 5 | 6 | 5 |
| TransD | 0.059 | 0.185 | 0.039 | 0.000 | 7 | 7 | 7 | 5 |
| ComplEx | 0.100 | 0.289 | 0.080 | 0.020 | 3 | 3 | 3 | 4 |
| DistMult | 0.089 | 0.203 | 0.080 | 0.033 | 4 | 4 | 3 | 2 |
| RotatE | 0.119 | 0.360 | 0.103 | 0.021 | 2 | 2 | 2 | 3 |
| BioBRIDGE | 0.326 | 0.653 | 0.382 | 0.177 | 1 | 1 | 1 | 1 |

Table 13: Cross-modality prediction performance for triple type: "Protein - interacts with - CC".

| Method | MRR | Hit@10 | Hit@3 | Hit@1 | MRR Rank | Hit@10 Rank | Hit@3 Rank | Hit@1 Rank |
|---|---|---|---|---|---|---|---|---|
| TransE | 0.044 | 0.133 | 0.026 | 0.000 | 8 | 8 | 7 | 5 |
| TransR | 0.048 | 0.142 | 0.023 | 0.000 | 7 | 7 | 8 | 5 |
| TransH | 0.057 | 0.176 | 0.029 | 0.000 | 5 | 5 | 5 | 5 |
| TransD | 0.053 | 0.162 | 0.026 | 0.000 | 6 | 6 | 6 | 5 |
| ComplEx | 0.099 | 0.267 | 0.081 | 0.023 | 3 | 3 | 3 | 3 |
| DistMult | 0.095 | 0.212 | 0.092 | 0.034 | 4 | 4 | 2 | 2 |
| RotatE | 0.107 | 0.330 | 0.079 | 0.018 | 2 | 2 | 4 | 4 |
| BioBRIDGE | 0.319 | 0.676 | 0.373 | 0.160 | 1 | 1 | 1 | 1 |

Table 14: Cross-modality prediction performance for triple type: "Drug - Indication - Disease".

| Method | MRR | Hit@10 | Hit@3 | Hit@1 | MRR Rank | Hit@10 Rank | Hit@3 Rank | Hit@1 Rank |
|---|---|---|---|---|---|---|---|---|
| TransE | 0.017 | 0.057 | 0.006 | 0.000 | 8 | 8 | 8 | 5 |
| TransR | 0.053 | 0.152 | 0.019 | 0.000 | 3 | 3 | 4 | 5 |
| TransH | 0.026 | 0.074 | 0.015 | 0.000 | 5 | 5 | 6 | 5 |
| TransD | 0.022 | 0.064 | 0.018 | 0.000 | 7 | 6 | 5 | 5 |
| ComplEx | 0.042 | 0.084 | 0.028 | 0.010 | 4 | 4 | 3 | 3 |
| DistMult | 0.025 | 0.063 | 0.010 | 0.004 | 6 | 7 | 7 | 4 |
| RotatE | 0.150 | 0.357 | 0.150 | 0.054 | 2 | 2 | 2 | 2 |
| BioBRIDGE | 0.189 | 0.380 | 0.176 | 0.071 | 1 | 1 | 1 | 1 |

Table 15: Cross-modality prediction performance for triple type: "Protein - Target - Drug".

| Method | MRR | Hit@10 | Hit@3 | Hit@1 | MRR Rank | Hit@10 Rank | Hit@3 Rank | Hit@1 Rank |
|---|---|---|---|---|---|---|---|---|
| TransE | 0.033 | 0.086 | 0.016 | 0.000 | 8 | 8 | 8 | 5 |
| TransR | 0.069 | 0.204 | 0.058 | 0.000 | 4 | 3 | 4 | 5 |
| TransH | 0.043 | 0.127 | 0.018 | 0.000 | 7 | 6 | 6 | 5 |
| TransD | 0.049 | 0.150 | 0.018 | 0.000 | 5 | 5 | 6 | 5 |
| ComplEx | 0.079 | 0.184 | 0.075 | 0.021 | 3 | 4 | 3 | 3 |
| DistMult | 0.044 | 0.104 | 0.035 | 0.008 | 6 | 7 | 5 | 4 |
| RotatE | 0.125 | 0.258 | 0.131 | 0.051 | 2 | 2 | 2 | 2 |
| BioBRIDGE | 0.172 | 0.299 | 0.181 | 0.107 | 1 | 1 | 1 | 1 |

Table 16: Cross-modality prediction performance for triple type: "Disease - Associated with - Protein".

| Method | MRR | Hit@10 | Hit@3 | Hit@1 | MRR Rank | Hit@10 Rank | Hit@3 Rank | Hit@1 Rank |
|---|---|---|---|---|---|---|---|---|
| TransE | 0.010 | 0.015 | 0.002 | 0.000 | 8 | 8 | 7 | 5 |
| TransR | 0.029 | 0.066 | 0.007 | 0.000 | 5 | 5 | 5 | 5 |
| TransH | 0.014 | 0.020 | 0.003 | 0.000 | 6 | 6 | 6 | 5 |
| TransD | 0.013 | 0.018 | 0.002 | 0.000 | 7 | 7 | 7 | 5 |
| ComplEx | 0.048 | 0.106 | 0.031 | 0.010 | 3 | 4 | 4 | 3 |
| DistMult | 0.047 | 0.113 | 0.035 | 0.010 | 4 | 3 | 3 | 3 |
| RotatE | 0.076 | 0.172 | 0.059 | 0.021 | 2 | 2 | 2 | 2 |
| BioBRIDGE | 0.081 | 0.184 | 0.071 | 0.025 | 1 | 1 | 1 | 1 |

Table 17: Cross-modality prediction performance for triple type: "Drug - Contraindication - Disease".

| Method | MRR | Hit@10 | Hit@3 | Hit@1 | MRR Rank | Hit@10 Rank | Hit@3 Rank | Hit@1 Rank |
|---|---|---|---|---|---|---|---|---|
| TransE | 0.010 | 0.015 | 0.002 | 0.000 | 8 | 8 | 7 | 5 |
| TransR | 0.029 | 0.066 | 0.007 | 0.000 | 5 | 5 | 5 | 5 |
| TransH | 0.014 | 0.020 | 0.003 | 0.000 | 6 | 6 | 6 | 5 |
| TransD | 0.013 | 0.018 | 0.002 | 0.000 | 7 | 7 | 7 | 5 |
| ComplEx | 0.048 | 0.106 | 0.031 | 0.010 | 3 | 4 | 4 | 3 |
| DistMult | 0.047 | 0.113 | 0.035 | 0.010 | 4 | 3 | 3 | 3 |
| RotatE | 0.076 | 0.172 | 0.059 | 0.021 | 2 | 2 | 2 | 2 |
| BioBRIDGE | 0.081 | 0.184 | 0.071 | 0.025 | 1 | 1 | 1 | 1 |

Table 18: The statistics of the built cross-specifies retrieval dataset.

| Mouse Protein - Mouse Phenotype | | | Mouse Protein - Human Phenotype | |
|---|---|---|---|---|
| Item | Train | Test | Item | Test |
| MGI | 6,069 | 6,068 | MGI | 1,615 |
| MPI | 28 | 28 | HPO | 353 |
| Pairs | 38,705 | 37,937 | Pairs | 3,515 |
| Mean MPI | 6.38 | 6.26 | Mean HPO | 2.18 |

# E   MOUSE PROTEIN AND PHENOTYPE DATASET

We retrieved the paired mouse protein and phenotype data from `https://www.informatics.jax.org/downloads/reports/HMD_HumanPhenotype.rpt`, yielding a total of 29,686 paired mouse protein ID (MGI), human protein ID, and mouse phenotype ID (MPI), namely `data1`. We then retrieved the detailed descriptions of mouse phenotypes from `https://www.informatics.jax.org/downloads/reports/MPheno_OBO.ontology`, yielding 14,201 MPI and their definition, namely `data2`.

To obtain the exact sequence structure of mouse proteins, we called UniProt `https://www.uniprot.org/id-mapping` mapping system, choosing "MGI" under "Organism-specific databases" as the source database, "UniproKB/Swiss-Prot" as the target database. This process maps MGI to the corresponding protein sequences, namely `data3`.

We only keep MGI that is with protein sequence referring to `data3` from `data1`. Also, we only keep MGI in `data1` that is linked to more than one MPI. Finally, we conduct a filter from `data1` to keep human proteins that are in the PrimeKG data and link to more than one human protein. The obtained dataset then has 14,024 MGI, 14,200 MPI, 1,582 Human Proteins, and 3,279 Human Phenotypes in total.

# F CASE STUDY: MULTIMODAL Q&A

Table 19: Additional case studies of multi-modal Q&A based on `BioBRIDGE`: Input a molecule and a question, retrieve the relevant diseases and proteins, and prompt LLM to answer. The references are drawn for the input molecule-based drug pages from DrugBank.

| Input Molecule | Question | Retrieved Disease | Retrieved Protein | Answer | Reference |
|---|---|---|---|---|---|
| C[C@H](C1=NOC (NC(=N)N2CCOCC2)=C1) C1=CC=C(C(F)=C1) C1=CC=CC=C1 | What are the associated conditions and pharmacodynamics of this drug? | osteoporosis / postmenopausal osteoporosis / psoriasis / allergic contact dermatitis / atopic eczema / rheumatoid arthritis / osteonecrosis / allergic asthma / systemic lupus erythematosus (disease) / congestive heart failure | MAPK14 / CDK2 / MAPK1 / MAPK8 / MAPK3 / EP300 / MAPK9 / CREBBP / GSK3B / MAPK10 | The associated diseases and the proteins this molecule targets suggest that this small molecule drug may have immunomodulatory, anti-inflammatory, and anti-arthritic effects. The drug may inhibit the activity of MAPK14, which plays a role in the production of pro-inflammatory cytokines. This inhibition may help reduce inflammation in conditions such as psoriasis, allergic contact dermatitis, atopic eczema, rheumatoid arthritis, and allergic asthma. The drug may also inhibit the activity of CDK2, which is involved in the regulation of the cell cycle and proliferation of immune cells. This inhibition may help suppress the immune system and reduce inflammation in systemic lupus erythematosus (disease). Additionally, the drug may inhibit the activity of other MAPK family members such as [...], which are involved in the regulation of immune responses and inflammation. The drug may also activate EP300 and CREBBP, which are transcriptional co-activators that can promote the expression of genes involved in anti-inflammatory responses. Overall, the main pharmacodynamics of this small molecule drug may be immunomodulatory, anti-inflammatory, and anti-arthritic effects, making it a potential treatment option for various conditions such as psoriasis, allergic contact dermatitis, atopic eczema | Rimacalib has been used in trials studying the treatment of Rheumatoid Arthritis (RA). [DrugBank: DB12571] |
| N[C@H](C(=O)N1 [C@H]2C[C@H]2C [C@H]1C#N)C12CC3CC (CC(O)(C3)C1)C2 | What are the associated conditions and pharmacodynamics of this drug? | diabetes mellitus (disease) / type 2 diabetes mellitus / coronary artery disease / bipolar disorder / bipolar II disorder / bipolar depression / bipolar I disorder / obesity disorder / dysthymic disorder / monogenic obesity / neurotic disorder / diabetic ketoacidosis | DPP4 / F2 / FAP / F9 / PRSS2 / PLAT / F10 / ACE / DPP6 / PRSS1 | Based on the associated diseases and the proteins this molecule targets, the main possible associated conditions of this small molecule drug are type 2 diabetes, obesity, coronary artery disease, neurotic disorder, dysthymic disorder, monogenic obesity, diabetic ketoacidosis, and bipolar disorders. The main pharmacodynamics of this small molecule drug are its inhibitory effects on dipeptidyl peptidase 4 (DPP4) and fibroblast activation protein alpha (FAP), which may help manage type 2 diabetes and obesity. The drug may also have some effects on the coagulation cascade, potentially reducing the risk of coronary artery disease and other coagulation-related disorders. Additionally, the drug may have some anti-inflammatory effects through its inhibition of FAP, which may contribute to its efficacy in managing some neurotic and mood disorders. | Saxagliptin is a dipeptidyl peptidase-4 (DPP-4) inhibitor antidiabetic for the treatment of type 2 diabetes. [DrugBank: DB06335] |
| CN/ C(NCCSCC1=CC=C (CN(C)C)O1)= C/[N+]([O-])=O | Which diseases can this molecule treat and why? | conjunctivitis (disease) / duodenal ulcer (disease) / pulmonary embolism (disease) / gastroenteritis / chronic duodenal ileus / peptic ulcer disease / intralobar congenital pulmonary sequestration / Echovirus infectious disease / trachoma / lower respiratory tract disease | NOS2 / NOS3 / GSTP1 / GSTA2 / GSTA1 / GSTM2 / GSTM1 / GSTA5 / GSTM4 / PTGS2 | Based on the target proteins and associated diseases listed, the main possible associated conditions of this small molecule drug are: 1. Conjunctivitis (disease) 2. Pulmonary embolism (disease) 3. Gastroenteritis 4. Chronic duodenal ileus 5. Peptic ulcer disease 6. Echovirus infectious disease The main pharmacodynamics of this small molecule drug would depend on its interaction with the target proteins. As it targets nitric oxide synthase 2 and 3, it may have vasodilation and anti-inflammatory effects. As it targets glutathione S-transferase pi 1, alpha 2, alpha 1, mu 2, mu 1, alpha 5, mu 4, and prostaglandin-endoperoxide synthase 2, it may have antioxidant and anti-inflammatory effects. Additionally, as it is a small molecule drug, it may have a good ability to cross cell membranes and reach its targets quickly. | Ranitidine is a histamine H2 antagonist used to treat duodenal ulcers, Zollinger-Ellison syndrome, gastric ulcers, GERD, and erosive esophagitis. [DrugBank: DB00863] |
| [H][C@@]12OC3=C (OC)C=CC4=C3[C@@] 11CCN(C)[C@]([H]) (C4)[C@]1([H]) C=C[C@@H]2O | What are the associated conditions and pharmacodynamics of this drug? | vasomotor rhinitis / allergic rhinitis / seasonal allergic rhinitis / perennial allergic rhinitis / common cold / atopic conjunctivitis / nasopharyngitis / papillary conjunctivitis / giant papillary conjunctivitis / vernal conjunctivitis / pharyngitis / angioedema / allergic asthma | OPRM1 / OPRK1 / OPRD1 / OPRL1 / HTR2C / HTR2A / DRD2 / CHRM2 / HTR2B / DRD3 | Based on the associated diseases and the proteins this molecule targets, the main possible associated conditions of this small molecule drug are pain, asthma, and allergic reactions. The main pharmacodynamics of this small molecule drug are its activity as an opioid receptor agonist, which can regulate the release of neurotransmitters involved in pain, mood, respiration, and immune function. It can also have anti-inflammatory and anti-allergic effects, although the exact mechanisms are not fully understood. Therefore, this drug may be effective in treating pain and asthma, as well as allergic conditions such as vasomotor rhinitis, allergic rhinitis, and allergic asthma. | Codeine is utilized as a central analgesic, sedative, hypnotic, antinociceptive, and antiperistaltic agent, and is also recommended in certain diseases with incessant coughing. [Drugbank: DB00318] |
| CN[C@@]1(C) [C@@H]2CC[C@@H] (C2)C1(C)C | What are the associated conditions and pharmacodynamics of this drug? | alcohol dependence / substance abuse / substance dependence / alcohol abuse / heroin dependence / cannabis dependence / cocaine abuse / cocaine dependence | SLC6A3 / HTR1A / CHRM2 / SLC6A2 / CHRM3 / SLC6A4 / DRD2 / CHRM4 / CHRM5 / HTR1B | The small molecule drug described targets multiple proteins involved in the regulation of various neurotransmitters and their receptor pathways. The associated diseases and conditions that this drug could possibly address include drug and alcohol dependence, substance abuse and dependence, and related disorders such as anxiety and depression. The main pharmacodynamics of this drug involve the modulation of the neurotransmitter systems of serotonin, dopamine, and acetylcholine, which are involved in the regulation of mood, emotional response, reward-seeking behaviors, and cognitive function. The specific actions of this drug on each of the target proteins are not explicitly stated in the given information, but based on the known functions of these proteins, this drug could potentially regulate the release, reuptake, and signaling of these neurotransmitters, leading to the alleviation of symptoms associated with the associated diseases and conditions. | Dexmecamylamine has been used in trials studying the basic science and treatment of Patients, Depression, Drug Abuse, Pharmacokinetics, and Renal Impairment, among others. [DrugBank: DB11807] |
| O[Al](O)OS(=O) (=O)OC[C@H]1O [C@@H](O[C@@]2 (COS(=O)(=O)O[Al] (O)O)O [C@H](OS(=O)(=O) O[Al](O)O)[C@@H] (OS(=O) (=O)O[Al](O)O) [C@@H]2OS(=O)(=O)O[Al] (O)O)[C@H](OS(=O)(=O) O[Al](O)O)[C@@H](OS(=O) (=O)O[Al](O)O)[C@@H] 1OS(=O)(=O)O[Al](O)O | What are the associated conditions and pharmacodynamics of this drug? | esophagitis (disease) / gallbladder disease / gallbladder disease 2 / gallbladder disease 3 / cirrhosis of liver / peptic ulcer disease / kidney disease / diarrheal disease / gastroesophageal reflux disease / asthma / pyoureter / inherited hemoglobinopathy / acute myocardial infarction / hemoglobinopathy / acquired hemoglobinopathy | TF / SOD2 / TGFB1 / IL6 / LTF / MMP2 / TFRC / ALB / VEGFA / CAT | Based on the information provided, it appears that the small molecule drug may have anti-inflammatory and antioxidant effects as it targets several proteins involved in inflammation and oxidative stress, including interleukin 6, superoxide dismutase 2, transforming growth factor beta 1, and vascular endothelial growth factor A. In terms of associated conditions, the drug may be effective in treating diseases that involve inflammation and oxidative stress, such as cirrhosis of the liver, peptic ulcer disease, diarrheal disease, and kidney disease. It may also have potential in treating gastroesophageal reflux disease and gallbladder diseases, which are also associated with inflammation and oxidative stress. However, it's important to note that these are just speculative associations based on the limited information provided. | Sucralfate aids in the healing of duodenal ulcers, relieving painful inflammation by creating a protective mechanical barrier between the lining or skin of the gastrointestinal tract and damaging substances 2. In addition, sucralfate acts to increase levels of growth factors locally, and also causes an increase in prostaglandins which are important in the healing of the mucosa (lining) of the gastrointestinal tract 2. [DrugBank: DB00364] |

Table 20: Case study of multi-modal generation based on `BioBRIDGE`: Input the target condition and the intended mechanism, retrieve the relevant proteins, and prompt LLM to generate the target small molecule drug.

| Target Condition | Target Effect | Retrieved Protein | Generated Molecule | Most Similar Drug |
|---|---|---|---|---|
| Depressive disorder | The drug treats depressive disorder by being a selective dopamine receptor antagonist, hence inhibiting dopaminergic hyperactivity. | DRD2 HTR1A OPRM1 HTR2A HTR2C IL1B BDNF CNR1 PENK TPH1 |  CN1CC[C@H]2CN(C[C@@H]3Cc4ccc(F)cc4[C@@H]3C[C@H]2C1 |  [DB00854] Levorphanol [H][C@@]12CCCC[C@@]11CCCN(C)[C@@H]2CC2=C1C=C(O)C=C2 Levorphanol is an opioid analgesic used to treat moderate to severe pain. |
| Depressive disorder | The drug treats depressive disorder by being a selective dopamine receptor antagonist, hence inhibiting dopaminergic hyperactivity. | DRD2 HTR1A OPRM1 HTR2A HTR2C IL1B BDNF CNR1 PENK TPH1 |  CN1CCC[C@H]1c2=CC=CC(C=C2)OC)OC3=CC=CC=C3 |  [DB12341] LY-2456302 CC1=CC=CC(C)=C1)[C@@H]1CCCN1CC1=CC=C(OC2=CC=C(C=C2F)C(N)=O)C=C1 Ly2456302 has been used in trials studying the health services research and basic science of Anxiety Disorders and Alcohol Dependence. |
| Breast cancer | We want to design a drug that can target one of the following proteins: [proteins..] to treat breast cancer. | BRCA2 NOTCH2 NOTCH1 TYMS EGFR NOTCH3 TP53 CDKN2A ESR2 NCOA1 |  CCCCCCCCCCCNC1=NC(=O)N([C@@H]2O[C@H](C)[C@@H](O)[C@H]2O)C=C1F |  [DB01101] Capecitabine CCCCCOC(=O)NC1=NC(=O)N(C=C1F)[C@@H]1O[C@H](C)[C@@H](O)[C@@H]1O Capecitabine is a nucleoside metabolic inhibitor indicated to treat different gastrointestinal, including pancreatic cancer, and breast cancer. |
| Breast Cancer | We want to design a drug that can target one of the following proteins: [proteins..] to treat breast cancer. | BRCA2 NOTCH2 NOTCH1 TYMS EGFR NOTCH3 TP53 CDKN2A ESR2 NCOA1 ERBB2 |  O=C1C2=C(O)C=CC([N+](=O)[O-])=C2C(=O)C2=C([N+](=O)[O-])C=CC([N+](=O)[O-])=C12 |  [DB01204] Mitoxantrone OCCNCCNC1=CC=C(NCCNCCO)C2=C1C(=O)C1=C(C(O)=CC=C1O)C2=O Mitoxantrone is a chemotherapeutic agent used for the treatment of secondary progressive, progressive relapsing, or worsening relapsing-remitting multiple sclerosis. |

Table 21: The baseline case study of multi-modal generation based on Galactica generator **without** the retrieval augmentation by `BioBRIDGE`: Input the target condition and the intended mechanism and prompt LLM to generate the target small molecule drug directly.

| Target Condition | Target Effect | Retrieved Protein | Generated Molecule | Most Similar Drug |
|---|---|---|---|---|
| malignant lymphoma | The inhibition of mitosis at metaphase of cancer cells via polychemotherapy. | N/A | CCOC(=O)C1(C2=CC=CC=C2)CCN(CCC2=CC=C(O)C(OCC3=CC=CC3)=C2)CC1 | [DB00454] Meperidine CCOC(=O)C1(CCN(C)CC1)C1=CC=CC=C1 Meperidine is an opioid agonist with analgesic and sedative properties used to manage severe pain. |
| malignant lymphoma | The inhibition of mitosis at metaphase of cancer cells via polychemotherapy. | N/A | C1=CC=C(C=C1)CSC2=NC=CC=N2 | [DB03255] Phenol OC1=CC=CC=C1 Phenol is an antiseptic and disinfectant used in a variety of settings. |
| Depressive disorder | The drug treats depressive disorder by being a selective dopamine receptor antagonist, hence inhibiting dopaminergic hyperactivity. | N/A | C=CCC1C=C(C)C2(C)CC(OC)C2OC(O)(C(=O)C(=O)N3CCCCC3C(=O)OC(C(C)=CC3CCC(O)(OC)C3)C)C(C)C(O)CC1=O)C(C)CC2OC | [DB00864] Tacrolimus CO[C@@H]1C[C@@H](CC[C@H]1O)\C=C\C(C)[C@@H]1OC(=O)[C@@H]2CCCCN2C(=O)C(=O)[C@]2(O)O[C@@H]([C@H](C[C@H]2C)OC[C@H](C[C@@H](C)\C(=C\[C@H](C)C[C@@H](O)[C@H]1C)OC Tacrolimus is a calcineurin inhibitor used to prevent organ transplant rejection and to treat moderate to severe atopic dermatitis. |
| Depressive disorder | The drug treats depressive disorder by being a selective dopamine receptor antagonist, hence inhibiting dopaminergic hyperactivity. | N/A | C1=CC=C(C2=C(C3=CC=NC=C3)C(C3=CC=NC=C3)=NO2)C=C1 | [DB00139] Succinic Acid OC(=O)CCC(O)=O A water-soluble, colorless crystal with an acid taste that is used as a chemical intermediate, in medicine, the manufacture of lacquers, and to make perfume esters. |
| Breast cancer | We want to design a drug to treat breast cancer. | N/A | CCCCCCCCCCCCCCCCCCCCCCCCOCC(O)CO | [DB06811] Polidocanol CCCCCCCCCCCCOCCOCCOCCOCCOCCOCCOCCOCCO Polidocanol is a sclerosing agent used for the treatment of uncomplicated spider veins and uncomplicated reticular veins, all less than 3 mm in diameter, in the lower extremity. |
| Breast Cancer | We want to design a drug to treat breast cancer. | N/A | CN(C)C1=CC=C(C(=O)C2=CC=CC=C2)C=C1 | [DB00139] Succinic Acid OC(=O)CCC(O)=O A water-soluble, colorless crystal with an acid taste that is used as a chemical intermediate, in medicine, the manufacture of lacquers, and to make perfume esters. |

# G   CASE STUDY: PROMPT

Listing 1: The prompt for prompting language models to answer user questions takes the drug molecule structure as the input.

```
prompt = """Drug molecule structure: [START_I_SMILES] {smiles}
    [END_I_SMILES]

Target proteins:
  {protein_names}

Associated diseases:
  {disease_names}

Consider the associated diseases and the proteins this molecule targets,
    {input_question}
"""
```

Listing 2: The prompt for prompting language models to generate drug molecule guided by user inputs in natural language.

```
prompt = """The drug may be targeting the proteins:

  {protein_names}

{text_guidance}

Generate the most possible SMILES structure of this drug.
"""
```

# H   ADDITIONAL RELATED WORKS

**Knowledge Graph Learning**. Knowledge graph embedding learning algorithms were proposed for KG completion tasks (Han et al., 2018), i.e., given the head node and relation, predict the most probable tail node. These approaches assume a fixed set of nodes and do not handle *inductive* prediction, i.e., prediction for nodes not seen in the training. More recent efforts utilize graph neural networks to handle inductive setup (Teru et al., 2020; Liu et al., 2021), which generate embeddings for new nodes by aggregating its neighbors' embeddings. Nevertheless, they still do not handle isolated new nodes with no link to known nodes. Also, the inherent node properties, e.g., the sequence structure of protein nodes, were usually neglected in the encoding. On the contrary, the proposed BioBRIDGE learns from a KG, representing new nodes with base FMs and transforming the embeddings via bridge modules without needing neighbor graph inputs.

**Multimodal Learning**. Early works in multi-modal learning focus on cross-modal retrieval, such as based on Canonical Correlation Analysis (CCA) to learn a common representation space for different modalities (Rasiwasia et al., 2010). In the era of deep learning, researchers turned to fuel deep models with large cross-modal data, such as text and image, to get better performances (Radford et al., 2021). Multimodal learning was also introduced to knowledge graph embeddings, such as in (Xie et al., 2019; Mousselly-Sergieh et al., 2018; Lu et al., 2022).z However, these methods primarily focus on visual and textual modalities, which may not directly suit our specific application needs. A critical limitation of these methods is their confinement to predicting nodes within the training KG during the testing phase because these methods all assume the KG is fixed. In contrast, BioBridge stands out by leveraging base unimodal Feature Models (FMs) to process inputs from any modality present in the KG. It can then convert these inputs into any desired target modality.

# I  TRAINING SETUPS

In this paper, we aim to prove that bridging uni-modal FMs is feasible. As such, we deliberately choose simple architectures, like transformers, to transform input embeddings and original InfoNCE loss to learn the transformation modules. All are kept to a minimum but necessary to make Bio-Bridge work.

We did experiments to check the sensitivity of BioBridge w.r.t. hyperparameters; we chose the batch size to be in $\{512, 1024, 4096\}$, and training epochs to be $\{10, 50, 100\}$. We found the performance was not significantly different for the tried batch size. The method turned out to be converging within 50 epochs, and training with more epochs does not lead to further improvement. As such, we keep the same set of hyperparameters for BioBridge across all experiments: batch size 4096, training epochs 50, and learning rate 1e-4.

In our experiments, we also did the ablations for the bridge module: Variant 1: in Eq. (1), removing the residual connection, i.e., using $h = \psi(z, c_i, c_j, r)$. Variant 2: in Eq. (1), using RotatE transformation, i.e., using $h = z \circ \psi(z, c_i, c_j, r)$. In our experiments, Variant 1 failed to converge; Variant 2 obtained a worse performance than the additive transformation in Eq. (1).

