# OpenReview forum: "BioBridge: Bridging Biomedical Foundation Models via Knowledge Graphs"
_ICLR.cc/2024/Conference — ICLR 2024 poster_

### Official Review · Reviewer_L5qe · 2023-10-30

**Soundness:** 4 excellent
**Presentation:** 3 good
**Contribution:** 4 excellent
**Rating:** 8
**Confidence:** 5

**Summary:**

The paper proposes to bridge multiple biological modalities through KG. The motivation is that most of the data are uni-modal, paired data is scarce and due to the combinatorial explosion, infeasible to collect and train a multi-modal model. The author learns a transformation layer among the uni-modal representations to achieve multi-modal learning.

**Strengths:**

- Interesting problem setting on bridging multi-modal FMs in biology. Makes intuitive sense.
- Interesting approach on the bridge module
- Lots of interesting applications and analysis

**Weaknesses:**

- The method proposes several new modules that lack motivation and seem ad hoc.
- The clarity of the technical methods of the paper can be improved

**Questions:**

- Looking at equation 1, the transformation seems dependent on the (1) target modality node type and (2) relation type and (3) individual node in the target modality. For example, given the same node drug, it could have one transformed representation for protein A, a different one for protein B, and a different one for disease, etc. Is that right? Could the authors describe the design choice of this? why not map all of them into a single unified space? This brings a separate question: how do you conduct a similarity search if the embedding is dependent on individual node?
- The transformation loss function (eq.2) makes intuitive sense, but there seem to be numerous other options to achieve similar goals such as self-supervised link prediction. Have the authors experimented with other transformation techniques? if so, could the authors provide any additional information? if not, could the authors describe the motivation on why choosing this particular approach?
- The author uses a transformer model to achieve the transformation. It seems unnecessary since it is just encoding 4 embeddings? Have the authors experimented with other simpler approaches?
- How are the negative samples created?
- For individual application, is the model trained on every possible bridge transformation or is the model different for each application and individual task?
- Cross-modality retrieval seems to be exactly link prediction. In that case, there are numerous approaches for GNN-based link prediction that is missing as a baseline and has shown much better performance compared to the KG embedding methods. Have the authors compared to any of the latest link prediction method?
- For semantic similarity task, I worry that there is data leakage. Since the protein node is connected to these GO terms in the PrimeKG, the embedding already implicitly knows the labels during training. Have the authors addressed this concern by conducting some holdout protein-go links?
- For PPI, since it is the same modality between the head and tail nodes, why do we need bridge module?
- For cross-species task, why is phenotype not available? it seems to be available in PrimeKG?
- For the multi-modal generation, it is an interesting application, but have the authors checked if the retrieved list is novel predictions or existing links? It will be great to compare with a baseline that is just retrieving the top K entities in the KG and show the difference on answers.



I am happy to raise score if the authors address my questions.

---

> ### Author Response · Authors · 2023-11-17
> **Response to Reviewer L5qe (1/2)**
>
> We appreciate your valuable input and have made efforts to address your concerns in our revised response.
>
> **1. Transformation module**
> > given the same node drug, it could have one transformed representation for protein A, a different one for protein B, and a different one for disease, etc. Is that right?
>
> Thanks for pointing out the phrases that cause confusion. Refer to Eq. (1), this drug will be transformed to the protein space with $\hat{h} = z + \psi(z, c_i, c_j, r_{ij})$, where $z$ is the drug’s raw embedding, $c_i$ indicates the **source modality** is drug, c_j indicates the **target modality** is protein, and $r_{ij}$ indicates the relation is `associate with`. In this sense, this drug is transformed to the protein space, yielding $\hat{h}$, which is not entangled with any individual protein embeddings, such that we can use $\hat{h}$ to match all the candidate proteins.
>
> **2. Transformation module**
> > why not map all of them into a single unified space?
>
> Our transformation module is relation-aware, i.e., it maps the same entity to different spaces considering the target relation. This allows the model to differentiate between drug-indicate-disease and drug-contraindicate-disease. A single unified space is not able to make such a distinction.
>
>
> **3. Choice of the bridge formula**
> > could the authors describe the motivation on why choosing this particular approach?
>
> In this paper, we aim to make a proof of concept about bridging multiple unimodal FMs, supervised with knowledge graphs. We chose InfoNCE loss since it was used by numerous multimodal learning algorithms, such as CLIP [1] and ImageBind [2]. In addition, we tried out some alternative transformation modules for Eq. (1). There are two variants:
> - Variant 1: in Eq. (1), removing the residual connection, i.e., using $h = \psi(z,c_i,c_j,r)$.
> - Variant 2: in Eq. (1), using RotatE transformation, i.e., using $h = z \circ \psi(z,c_i,c_j,r)$.
>
> In our experiments, Variant 1 failed to converge; Variant 2 obtained a worse performance than the additive transformation in Eq. (1).
>
>
> [1] Radford A, Kim J W, Hallacy C, et al. Learning transferable visual models from natural language supervision[C]//International conference on machine learning. PMLR, 2021: 8748-8763.
>
> [2] Girdhar R, El-Nouby A, Liu Z, et al. Imagebind: One embedding space to bind them all[C]//Proceedings of the IEEE/CVF Conference on Computer Vision and Pattern Recognition. 2023: 15180-15190.
>
> **4. Choice of transformers in the bridge module**
> > The author uses a transformer model to achieve the transformation. It seems unnecessary.
>
> We build the bridge module with transformer layers for the following reasons:
> - (1) transformers ensure the input and output embeddings are with the same dimension, which provides a convenient way to achieve the transformation across modalities;
> - (2) transformer offers an effective way of feature interactions between the inputs;
> - (3) transformers were proven with high learning capacity, as they are used in many large language models, and we choose transformers by default to keep the architecture minimum viable, as a proof of the validity of the bridging.
>
>
> **5. Creation of the negative samples**
> > How are the negative samples created?
>
> We thank the reviewer for the suggestion. We randomly sampled from the same triple types. E.g. (h, r, t) (h, r, t_neg), t_neg are sampled from e_t that do not have the same relation r with h. Please refer to the description we moved to Sec 3.2 Page 4 “We corrupt $t_{ij}$ by replacing…”.
>
> **6. Data leakage in semantic similarity task**
> > For semantic similarity task, I worry that there is data leakage
>
> Thank you for this question. In order to explicitly avoid data leakage, we have removed all proteins in the testing set from the training KG. In this case, all testing protein-go links were not visible to the BioBridge model. We reason that the improvement led by projecting protein embedding to the GO term space stems from the discriminative patterns of GO terms formed by PubMedBERT for associated proteins. This discriminative pattern in GO term space is more pronounced than the original protein embedding space, which only exploits proteins’ internal information.
>
>
> **7. Bridge Module for PPI**
> > since it is the same modality between the head and tail nodes, why do we need bridge module?
>
> There are 7 types of relations representing different types of protein-protein interactions (PPIs): reaction, binding, post-translational modifications (ptmod), activation, inhibition, catalysis, and expression. So, it is still useful if we can perform a bridge within the same modality considering different relations. We also proved bridging proteins with PPI relation leads to better performance in PPI prediction tasks than the original embedding by ESM2 models, as shown in Sec 4.3 Table 3 Page 7, where the transformed protein embeddings considering PPI relations induce better performance than the protein embedding baselines.

---

> ### Author Response · Authors · 2023-11-17
> **Response to Reviewer L5qe (2/2)**
>
> **8. Phenotype in PrimeKG**
> > For cross-species task, why is phenotype not available? it seems to be available in PrimeKG?
>
> Thanks for pointing it out. This paper aims to make a proof of concept in bridging uni-modal FMs, so it simplifies the KG in the experiment.  Nonetheless, protein, disease, drug, and gene ontology terms are the main biomedical entities covering the majority of real-world biomedical tasks (e.g., drug discovery, repurposing, PPI, protein function prediction, drug target identification, drug-target interaction prediction). Please refer to Table 8 for the data statistics. We also added the explanation to Appendix B Page 15 in the new version.
>
> **9. Multimodal generation**
> > For the multi-modal generation, it is an interesting application, but have the authors checked if the retrieved list is novel predictions or existing links? It will be great to compare with a baseline that is just retrieving the top K entities in the KG and show the difference on answers.
>
>
> Thanks for this question. We added experiments to verify for the molecule generation cases. For the retrieved top-20 relevant proteins by BioBridge, we compared them with the existing KG links in the training graph to get the proteins in KG and not in KG:
>
>
> | Disease           | Linked in KG                                        | Not-linked in KG                                    |
> |-------------------|------------------------------------------------------|------------------------------------------------------|
> | malignant lymphoma| MTHFR, BCL2, TYMS, CSF3, CREBBP, FAS, BHMT, BCL6, BCL10| CAT, CASP8, CDKN2A, CDKN2A, TP53, SOD2, PRDM1, STAT3, PIK3CA, IL2, MALT1, MYD88 |
> | Depressive Disorder| DRD2, OPRM1, HTR2A, HTR2C, IL1B, CNR1, PENK, TPH1, GRIA2, POMC, HTR1B, HTR4, DRD3, GRIA1, IL6 | HTR1A, BDNF, DRD1, GRIN1, GRIK2 |
> | Breast Cancer    | BRCA2, NOTCH1, TYMS, EGFR, NOTCH3, TP53, CDKN2A, CDKN2A, ESR2, NCOA1, ERBB2, TERT, KDR, FLT1, HIF1A, BRCA1, PTGS2, JAG1 | NOTCH2, NCOA3, RAD51 |
>
>
> It can be seen
> - (1) BioBridge is able to retrieve many true KG (on average ~70%) linked entities given the input disease descriptions and
> - (2) BioBridge can also make novel predictions (on average ~30%) that are not in the training KG.
>
> The assumption that selecting the top K entities as context will be at least as effective as using the BioBridge retrieval method seems logical because the entities linked through the knowledge graph (KG) provide completely accurate contexts. However, this KG-linked retrieval approach doesn't suit our situation, as our inputs aren't associated with a precise KG node index. Instead, our inputs are SMILES strings or protein sequences, which we assume may not be in the KG.

---

> ### Author Response · Authors · 2023-11-22
> **Kindly reminder of the end of the rebuttal period**
>
> Dear Reviewer L5qe,
>
> As the discussion period is coming to an end tomorrow, we kindly ask you to review our response to your comments and let us know if you have any further queries. Alternatively, if you could raise the score of the paper, we would be extremely grateful. We eagerly anticipate your response and are committed to addressing any remaining concerns before the discussion period concludes.
>
> Best regards,
>
> Authors

---

> ### Comment · Reviewer_L5qe · 2023-11-22
>
> Thanks for the detailed answers! It has addressed my main concerns. I am raising the score to 8.

---

### Official Review · Reviewer_xFRj · 2023-10-31

**Soundness:** 3 good
**Presentation:** 3 good
**Contribution:** 3 good
**Rating:** 6
**Confidence:** 3

**Summary:**

This paper proposes BioBRIDGE to bridge unimodal FMs, keeping each FM fixed. BioBRIDGE utilizes a multi-modal knowledge graph to learn cross-modal relationships of entities. Compared to multi-modal FMs and ImageBind, BioBRIDGE is parameter efficient. The experiments on cross-modal retrieval tasks demonstrate the effectivness of BioBRIDGE. The paper also shows that BioBRIDGE has good generalization ability.

**Strengths:**

1. The idea of bridging several unimodal FMs is novel.
2. The paper is well written and easy to follow.
3. Compared to existing studies, BioBRIDGE keeps all unimodal FMs fixed and thus is parameter efficient.

**Weaknesses:**

1. The reasons of using contrastive learning is not clear. It would be better to provide further explanation and experimental supports.
2. The baselines in Section 4.1 are several years ago. No recent studies are included.

**Questions:**

1. What are the contributions of the contrastive learning?
2. How about the influence of the knowledge graph on this method? From your perspective, what could be the challenges if BioBRIDGE is adapted to other domains?
3. For the cross-modality retrieval tasks, are the baselines trained on single modality or multiple modalities? Why not include multi-modal embedding methods published recently?

    a. Lu X, Wang L, Jiang Z, et al. MMKRL: A robust embedding approach for multi-modal knowledge graph representation learning[J]. Applied Intelligence, 2022: 1-18.

    b. Cao X, Shi Y, Wang J, et al. Cross-modal knowledge graph contrastive learning for machine learning method recommendation[C]//Proceedings of the 30th ACM International Conference on Multimedia. 2022: 3694-3702.

    c. Zhu J, Huang C, De Meo P. DFMKE: A dual fusion multi-modal knowledge graph embedding framework for entity alignment[J]. Information Fusion, 2023, 90: 111-119.
4. Though it is easier to collect unimodal data than to collect paired data from two modalities, could the model trained on the paired data perform better or competitively to model trained on the large unimodal data?

---

> ### Author Response · Authors · 2023-11-17
> **Response to Reviewer xFRj**
>
> We appreciate your valuable input and have made efforts to address your concerns in our revised response.
>
> **1. Reason for using contrastive learning**
> > The reasons of using contrastive learning is not clear … contributions of contrastive learning
>
> Contrastive learning has been widely used in multimodal representation learning, such as in CLIP and ImageBind. Here, we leverage contrastive learning to push the transformed head embeddings close to the target tail node embeddings by optimizing the parameters of the transformation model. We do not intend to propose a new contrastive learning algorithm but choose InfoNCE loss to validate the idea of bridging.
>
> **2. Baselines in cross-modal prediction**
> > Are the baselines trained on single-modality or multiple-modalities
>
> In cross-modal prediction tasks, various methods are trained using the same Knowledge Graph (KG) incorporating a range of node types. This approach is evident in a series of multimodal graph learning research, including studies like IKRL [1], TBKGE [2], TransAE [3], and MMKRL [4]. However, these methods primarily focus on visual and textual modalities, which may not directly suit our specific application needs. A critical limitation of these methods is their confinement to predicting nodes within the training KG during the testing phase because these methods all assume the KG is fixed. In contrast, BioBridge stands out by leveraging base unimodal Feature Models (FMs) to process inputs from any modality present in the KG. It can then convert these inputs into any desired target modality. We have added this discussion to Appendix H Page 22, in the new version.
>
> [1] Xie R, Heinrich S, Liu Z, et al. Integrating image-based and knowledge-based representation learning[J]. IEEE Transactions on Cognitive and Developmental Systems, 2019, 12(2): 169-178.
>
> [2] Mousselly-Sergieh H, Botschen T, Gurevych I, et al. A multimodal translation-based approach for knowledge graph representation learning[C]//Proceedings of the Seventh Joint Conference on Lexical and Computational Semantics. 2018: 225-234.
>
> [3] Wang Z, Li L, Li Q, et al. Multimodal data enhanced representation learning for knowledge graphs[C]//2019 International Joint Conference on Neural Networks (IJCNN). IEEE, 2019: 1-8.
>
> [4] Lu X, Wang L, Jiang Z, et al. MMKRL: A robust embedding approach for multi-modal knowledge graph representation learning[J]. Applied Intelligence, 2022: 1-18.
>
>
> **3. Generalizability of BioBridge**
> > influence of the KG; what could be the challenges if BioBRIDGE is adapted to other domains?
>
> In our experiments, the used PrimeKG contains six modalities, and most of the relations are actually for two nodes from the same modality, such as protein-protein and molecule-molecule. The proposed method can work better for cross-modal transformation if more triples across modalities are available. BioBridge proposes a general framework for bridging multiple modalities by learning from KGs, focusing on the biomedical domain. The task “multimodal QA” in our multimodal generation case study (Sec 5) is analogous to image captioning/visual QA: given a molecule (image) and a question, answer the question. BioBridge was able to bridge multiple biomedical modalities; it will also work for other modalities. We encourage further efforts in exploring bridging other modalities like vision, text, and audio if the corresponding KGs are available.
>
>
>
> **4. The model trained on paired v.s. uni-modal models**
> > could the model trained on the paired data perform better or competitively to model trained on the large unimodal data?
>
> Yes. In Table 2, OntoProtein, KeAP are trained on paired data (protein, text), we found KeAP performed slightly better than ESM-1B, which is trained on large unimodal data. However, paired data greater than two modalities are rare and hard to collect.

---

> ### Author Response · Authors · 2023-11-22
> **Kindly reminder of the end of the rebuttal period**
>
> Dear Reviewer xFRj,
>
> As the discussion period is coming to an end tomorrow, we kindly ask you to review our response to your comments and let us know if you have any further queries. Alternatively, if you could raise the score of the paper, we would be extremely grateful. We eagerly anticipate your response and are committed to addressing any remaining concerns before the discussion period concludes.
>
> Best regards,
>
> Authors

---

> > ### Comment · Reviewer_xFRj · 2023-11-22
> > **Reply to the response**
> >
> > Dear authors,
> >
> > Thanks for the clarification. There are no further questions. The baselines employed in Section 4.1 are not SOTA and thus not convincing enough. I would like to keep my score.
> >
> > Best.

---

### Official Review · Reviewer_VYFb · 2023-11-02

**Soundness:** 3 good
**Presentation:** 3 good
**Contribution:** 2 fair
**Rating:** 6
**Confidence:** 4

**Summary:**

This paper introduces BioBridge, a novel framework for training across modalities (entities) that bridges independently trained uni-task models to establish cross-task abilities in biomedical domains. BioBridge employs contrastive learning to align entity representations and to facilitate learning of transformations between them. The resulting model demonstrates strong performance relative to several baseline knowledge graph (KG) embedding methods in retrieval tasks and highlights potential applications in the guided discovery of new drugs. Overall, BioBridge represents a promising approach for the integration of biomedical data resources and the enhancement of performance in various downstream tasks.

**Strengths:**

- This paper proposes a novel concept for learning across modalities via the bridging of knowledge graphs.
- The authors have conducted extensive experiments on various types of entity mapping and numerous approaches to tail entity prediction.
- With only the bridge module requiring updates during training, and all base feature models (FMs) remaining fixed, the proposed method is computationally efficient.
- Overall, the paper is well-written, with clear explanations of the methodology and empirical results.

**Weaknesses:**

- The term "multimodal" as mentioned in this paper is confined to different types of biomedical entities. While the authors compare their work with "ImageBind," the experimental section lacks tasks that bridge text and image modalities, which are more complex and crucial for multimodal foundation models.
- The learning process is guided by knowledge graphs, limiting the scope of "modalities" to those represented within biomedical knowledge graphs. Therefore, instead of the broad term "biomedical foundation model," it would be more accurate to describe it as a "biomedical knowledge graph foundation model."
- The paper does not present ablation studies, such as evaluations of the contrastive learning objectives or hyper-parameter tuning.
- The case study focusing on molecule generation is intriguing. Quantitative assessments of generation performance would be beneficial, for instance, by making direct comparisons with general-domain foundation models, or by offering more qualitative examples to demonstrate the efficacy of the proposed framework.
- The baseline comparisons are predominantly with knowledge graph link prediction methods. It is unclear whether the observed advantages stem from the effective transformation learning of the proposed method or from the knowledge supervision of the biomedical knowledge graph.
- This paper omits a discussion on limitations and potential failure modes. The authors are strongly encouraged to offer deeper insights into the generalizability of BioBridge.

**Questions:**

- Can the proposed method be applied to other types of downstream tasks, such as image captioning and visual question answering? If applicable, could the authors provide empirical results and case studies?
- Could the author offer a more detailed explanation of the compared baselines? Specifically, how are they trained, and can they also learn from an external biomedical knowledge graph to ensure a fair comparison?
- Would employing different contrastive learning objectives, such as SimCLR or MoCo, in place of InfoNCE, impact the performance?
- Also, please refer to weaknesses for other concerns.

---

> ### Author Response · Authors · 2023-11-17
> **Response to Reviewer VYFb (1/3)**
>
> We appreciate your valuable input and have made efforts to address your concerns in our revised response.
>
> **1. The choice of modalities in the task**
> > the experimental section lacks tasks that bridge text and image modalities, which are more complex and crucial for multimodal foundation models.
>
> We agree that texts and images are two crucial modalities for multimodal learning, including in the biomedical domain. In this paper, we do include text and prove the feasibility of bridging text with other modalities, such as molecules and proteins, via bridging. The proposed method leverages modalities mentioned in the used KG. As such, it can be extended to image modality if we have a KG with image nodes. For instance, medical images can be connected to such KG via patient-image-disease relations. However, no multi-condition/disease medical imaging datasets are publicly available. Although there are a few single-disease medical image datasets across patients (e.g., MIMIC-CXR), it is not meaningful to connect all of the images to just one disease node in a KG.
>
> **2. Biomedical foundation models v.s. biomedical knowledge graph foundation models**
> > learning process is guided by knowledge graphs, …, it would be more accurate to describe it as a "biomedical knowledge graph foundation model.”
>
> We thank the reviewer for the suggestion. In our paper, KG is leveraged as the supervision for the bridge module to apply transformation across biomedical uni-modal foundation models (FMs). As the KG is not involved in the testing phase since we encode inputs using the base FMs and apply transformation via the bridge module,  we believe that “bridging biomedical foundation models” is an accurate description of the proposed method. We have added the discussion in Sec 3.1, Page 3, to clarify further. Please check the edits in red in the new version.
>
> **3. Selected baselines**
> > baseline comparisons are predominantly with knowledge graph link prediction
>
> While KG line prediction algorithms are used as the baseline in the first experiment, we included the state-of-the-art algorithms for the other experiments. For example, OntoProtein, KeAP, and ESM2 are shown in Sections 4.2 and 4.3. In 4.2, our method yields a 2x better performance in semantic similarity inference than the SOTA baselines such as ESM2, OntoProtein, and KeAP; In 4.3, our method is better than KeAP and ESM2 in protein-protein interaction prediction.
>
> **4. Source of the improvement**
> > it is unclear whether the observed advantages stem from the effective transformation learning of the proposed method or from the knowledge supervision of the biomedical knowledge graph.
>
> The improvement comes from two sources: Firstly, it is built upon uni-modal FMs that are trained on extensive datasets. Secondly, it utilizes knowledge graphs to interconnect these uni-modal FMs, thereby elevating their predictive capabilities. For example, in the studies detailed in Section 4.2 on Page 6, BioBridge outperforms both uni-modal protein FMs like ProtBERT and ESM, and knowledge graph-guided protein models such as KeAP and OntoProtein. This superior performance is attributed to BioBridge's ability to convert the base protein FM embeddings into the embedding space of Gene Ontology (GO) terms. Similarly, in the experiments described in Section 4.3 on Page 7, BioBridge shows better results than the protein FM ESM2-3B and the knowledge-informed model KeAP. This is due to BioBridge enhancing its base protein FM by incorporating protein-protein interaction (PPI) relations derived from the knowledge graph.
>
> **5. Apply to other modalities**
> > Can the proposed method be applied to other types of downstream tasks, such as image captioning and visual question answering?
>
> BioBridge proposes a general framework for bridging multiple modalities by learning from KGs, focusing on the biomedical domain. The task “multimodal QA” in our multimodal generation case study (Sec 5) is analogous to image captioning/visual QA: given a molecule (image) and a question, answer the question. BioBridge was able to bridge multiple biomedical modalities; it will also work for other modalities. We encourage further efforts in exploring bridging other modalities like vision, text, and audio if the corresponding KGs are available.

---

> ### Author Response · Authors · 2023-11-17
> **Response to Reviewer VYFb (2/3)**
>
> **6. A fair comparison with the baselines**
> > More detailed explanation of the compared baselines … can they also learn from biomedical KGs to ensure a fair comparison
>
> Thank you for emphasizing the importance of maintaining fair comparisons in experimental setups. We have thoroughly reviewed all the experiments.
>
> In Section 4.1, detailed in Table 1 on Page 6, BioBridge is compared with various KG embedding methods. In this comparison, all methods are trained on the same knowledge graph. However, BioBridge has the advantage of incorporating unimodal protein, molecule, and text feature modules.
>
> In Section 4.2, as shown in Table 2 on Page 6, the comparison includes two categories of baselines: protein embedding models such as ProtBERT and ESM, and knowledge-guided protein embedding models like OntoProtein and KeAP. For example, OntoProtein uses the ProteinKG25 dataset, which contains approximately 4.8 million protein-GO triples. The knowledge-guided models were observed to outperform the protein embedding models, attributed to the knowledge-based supervision. Despite accessing fewer protein-GO triples (about 0.29 million) in PrimeKG compared to KeAP's use of ProteinKG25, BioBridge still demonstrates superior performance.
>
> In Section 4.3, covered in Table 3 on Page 7, the comparison involves ESM, which are protein models without external knowledge, and KeAP trained with ProteinKG25. In these setups, too, BioBridge achieves better performance than both baselines.
>
> From this comprehensive review, it is evident that BioBridge consistently outperforms these baselines in fair and comparable experimental conditions.
>
>
> **7. Contrastive learning objectives**
> > Would employing different contrastive learning objectives, …, impact the performance
>
> Thank you for bringing this up. Yes, this is a critical point. The choice of different contrastive learning objectives when using bridge modules is an important area of future work (which we are exploring), both theoretically and empirically. In this work, we aim to validate the concept of bridging unimodal FMs with KGs. Hence, we chose InfoNCE loss, which was used in numerous multimodal learning methods, such as CLIP and ImageBind.
>
>
> **8. ablation study**
> > the paper does not present ablation studies
>
> We agree that ablation is important. In this paper, we aim to prove that bridging uni-modal FMs is feasible. As such, we deliberately choose simple architectures, like transformers, to transform input embeddings and original InfoNCE loss to learn the transformation modules. All are kept to a minimum but necessary to make BioBridge work.
>
> We did experiments to check the sensitivity of BioBridge w.r.t. hyperparameters; we chose the batch size to be in $\{512, 1024, 4096\}$, and training epochs to be $\{10, 50, 100\}$. We found the performance was not significantly different for the tried batch size. The method turned out to be converging within 50 epochs, and training with more epochs does not lead to further improvement. As such, we keep the same set of hyperparameters for BioBridge across all experiments: batch size 4096, training epochs 50, and learning rate 1e-4.
>
> In our experiments, we also did the ablations for the bridge module:
> - Variant 1: in Eq. (1), removing the residual connection, i.e., using $h = \psi(z,c_i,c_j,r)$.
> - Variant 2: in Eq. (1), using RotatE transformation, i.e., using $h = z \circ \psi(z,c_i,c_j,r)$.
>
> In our experiments, Variant 1 failed to converge; Variant 2 obtained a worse performance than the additive transformation in Eq. (1).
>
> We have added these ablations and analyses to Appendix I on Page 22.

---

> ### Author Response · Authors · 2023-11-17
> **Response to Reviewer VYFb (3/3)**
>
> **9. molecule generation**
> > quantitative assessments of generation performance would be beneficial
>
> We agree that a quantitative assessment of generation performance supported by BioBridge is interesting. In our testing case, we used biobridge as a retriever to enable conditional molecule generation guided by text prompts. To the best of our knowledge, we did not find testing sets suitable for our evaluation setups. Meanwhile, the generation performance is also influenced by the generator used; in our case, we use Galactica. This is not the core contribution of this paper, and we would like to leave it in for future research to build on with many applications. Below is the comparison of Galactica generating from scratch and augmented by BioBridge retrieval:
>
> - Question: Lymphoma is a broad term for cancer that begins in cells of the lymph system. The drug treats malignant lymphoma by the inhibition of mitosis at metaphase of cancer cells, via polychemotherapy. Generate the most possible SMILES structure of this drug.
>   - Galactica with BioBridge: C#CC(C)(C)NC(=O)C1=CC(Cl)=CC(Cl)=C1, most similar drug: Procarbazine DB01168, which is an antineoplastic agent indicated for the treatment of stage III and stage IV Hodgkin's disease in combination with other chemotherapeutic agents.
>   - Galactica without retrieval: CCOC(=O)C1(C2=CC=CC=C2)CCN(CCC2=CC=C(O)C(OCC3=CC=CC=C3)=C2)CC1, most similar drug: Meperidine DB00454 is an opioid agonist with analgesic and sedative properties used to manage severe pain.
>
> - Question: Depressive disorder is a common mental disorder. It involves a depressed mood or loss of pleasure or interest in activities for long periods of time. The drug treats depressive disorder by being a selective dopamine receptor antagonist, hence inhibiting dopaminergic hyperactivity. Generate the most possible SMILES structure of this drug that treats depressive disorder.
>   - Galactica with BioBridge: CN1CC[C@H]2CN(C[C@@H]3Cc4ccc(F)cc4[C @@H]3[C@H]2C1, most similar drug: Levorphanol DBH00854 which is an opioid analgesic used to treat moderate to severe pain.
>   - Galactica without retrieval: C1=CC=C(C2=C(C3=CC=NC=C3)C(C3=CC=NC=C3)=NO2)C=C1, most similar drug: Irospan 24/6 Kit DB00139 which is a water-soluble, colorless crystal with an acid taste that is used as a chemical intermediate.
>
> It can be seen that without retrieval with BioBridge, Galactica tends to generate irrelevant molecules. We have added more examples to Table 21, Page 21, in the new version. We also added the explanation of the most similar drugs to Table 6, Page 9.

---

> > ### Comment · Reviewer_VYFb · 2023-11-22
> >
> > Thank the authors for their further explanations. My main concerns are addressed. I have raised my score.

---

> ### Author Response · Authors · 2023-11-22
> **Kindly reminder of the end of the rebuttal period**
>
> Dear Reviewer VYFb,
>
> As the discussion period is coming to an end tomorrow, we kindly ask you to review our response to your comments and let us know if you have any further queries. Alternatively, if you could raise the score of the paper, we would be extremely grateful. We eagerly anticipate your response and are committed to addressing any remaining concerns before the discussion period concludes.
>
> Best regards,
>
> Authors

---

### Official Review · Reviewer_cgGU · 2023-11-02

**Soundness:** 3 good
**Presentation:** 3 good
**Contribution:** 3 good
**Rating:** 8
**Confidence:** 4

**Summary:**

This submission tackles the problem of learning large multi modal ML models without requiring the pairwise cross modal dataset (as it is infeasible in where no of models >2). unlike recent work that aligns all modalities to a single modality, this submission takes a different approach of learning cross modal alignment transformation in embedding space while keeping the underlying unimodal fixed/frozen. Given Given an input embedding that was encoded by a unimodal model, the proposed submission transforms it to the embedding space of the target modality accounting for their relations. the cross modal alignment transformation is parametrized by a vanilla transformer module and learned with a contrastive loss. The proposed method is evaluated on several benchmarks/tasks such as protein protein interaction, protein-phenotype matching, cross modal retrieval where it outperforms several baselines.

**Strengths:**

- the problem of aligning large unimodal models efficiently is very relevant in general and even more so when the modalities are proteins, drugs and diseases (focus of this submission) as it opens up plethora of clinical applications.

- the problem is well motivated in introduction and contextualised. The paper is clearly written and easy to follow except one section (see below).

- the idea of aligning the different embedding space of unimodal pretrained models with a cross modal transformation is sensible and simple.

- experimental validation: evaluation of proposed s convincing on several benchmarks where the proposed method outperforms several baselines and in some cases is the only applicable solution. evaluation and applicability of the proposed method on cross modal retrival and gene-phenotype matching are quite interesting.

**Weaknesses:**

- Presentation of Related work : Currently the submission only has one paragraph on knowledge graph learning and barely describes the embedding alignment literature e.g. in the context of cross modal retrieval (one of the application of proposed method), one can also mention deep CCA literature as Canonical correlation analysis (CCA) is the core of many cross modal retrieval methods.

- Presentation of Methodology:  the submission should motivate the solution somewhat intuitively. Section 3.2 on encoding and transformation is very to the point and concise. the proposed methodology could be motivated better e..g by contextualizing wrt some prior work on KGE literature. Although there is a paragraph in the related work on KGE completion, the proposed method is not contexualized. Similarly, It is not very clear to me what parameters are optimized with contrastive loss since the submission keeps the pretrained model frozen.

- Parameterization of alignment transformation as a transformer : the submission should also somehow motivate this choice wrt other options starting with simplest one such as a MLP or a vanilla autoencoder.

**Questions:**

Could the submission place the main assumptions and theorem from appendix into the main text? The appendix could include the proof.  I imagine this will make main paper more self contained and detailed. In the current form, methodology section is quite short.

---

> ### Author Response · Authors · 2023-11-17
> **Response to Reviewer cgGU**
>
> We appreciate your valuable input and have made efforts to address your concerns in our revised response.
>
>
> **1. Related work in embedding alignment literature, like CCA**
> > one can also mention deep CCA literature, …, core of many cross-modal retrieval methods
>
> We acknowledge the reviewer’s notice of prior literature on cross-modality retrieval, such as canonical correlation analysis (CCA). We have added more discussion to the prior cross-modal retrieval literature to the new appendix due to the page limit. Since we are working on biomedical tasks, our previous discussion was mostly on multimodal learning for biomedical modalities, as described in the 2nd paragraph of Sec 2. Their main weaknesses are they only focus on two modalities and require large paired datasets.
>
> **2. More intuitive motivation for the proposed method**
> > the submission should motivate the solution somewhat intuitively
>
> We have extended the discussion in related work about KGE embeddings to the method section to provide a sensible motivation for the proposed method. Although traditional KG embedding (KGE) methods also enable cross-modal prediction via link prediction, they do not generally extrapolate to nodes that are not in the training KG. Instead, BioBridge learns from the triplets to bridge the modalities by transforming the head-modality FM embeddings to the tail's space. In the testing phase, KG is no longer needed for the inputs. Also, we have made it explicit that the parameters of the transformation module and the projection head are optimized in training, with all unimodal FMs frozen. Please find the edits in red in the new version in Sec 3.1: “Although traditional KG embedding (KGE) methods also enable cross-modal prediction via link prediction, they do not extrapolate to nodes not in the training KG….”.
>
> **3. Choice of the transformation module.**
> > the submission should also somehow motivate this choice wrt other options starting with simplest one such as a MLP or a vanilla autoencoder.
>
> Thanks to the reviewer for bringing this important point up. This work aims to validate the concept of bridging modalities via KGs. We picked transformers because they were proven with sufficient capacity to learn representations from large data, as shown in numerous papers in training transformer-based models on large-scale data, e.g., BERT [1], GPT-3 [2], ViT [3]. Also, they keep the input and output embeddings in the same dimension, which is convenient for cross-modal embedding transformations. While this paper focuses on a proof of concept of bridging unimodal FMs, we acknowledge that it is interesting to explore diverse architectures for the transformation module and will experiment with them in the future.
>
>
> [1] Devlin J, Chang M W, Lee K, et al. Bert: Pre-training of deep bidirectional transformers for language understanding[J]. arXiv preprint arXiv:1810.04805, 2018.
>
> [2] Brown T, Mann B, Ryder N, et al. Language models are few-shot learners[J]. Advances in neural information processing systems, 2020, 33: 1877-1901.
>
> [3] Dosovitskiy A, Beyer L, Kolesnikov A, et al. An Image is Worth 16x16 Words: Transformers for Image Recognition at Scale[C]//International Conference on Learning Representations. 2020.
>
>
> **4. Moving the theorem to the main text**
>
> Sure, we have moved the theorem to Sec 3.4 on Page 5 in the updated version.

---

> ### Author Response · Authors · 2023-11-22
> **Kindly reminder the end of rebuttal period**
>
> Dear Reviewer cgGU,
>
> As the discussion period is coming to an end tomorrow, we kindly ask you to review our response to your comments and let us know if you have any further queries. Alternatively, if you could raise the score of the paper, we would be extremely grateful. We eagerly anticipate your response and are committed to addressing any remaining concerns before the discussion period concludes.
>
> Best regards,
>
> Authors

---

### Author Response · Authors · 2023-11-17
**General Response: New manuscript revision**

We would like to signal the upload of a new manuscript revision. This includes the changes anticipated in the previous general comment and throughout the responses to each single reviewer.

The revision includes the following main additions (reported in the paper appendix):

- Expand the related work section and add the discussion of more related papers in Appendix H on Page 22
- Clarify the motivation of the proposed method in Sec 3.1 on Page 3
- Add the theorem for proving the existence of bridge module and learnability to Sec 3.4 on Page 5
- Add the results of Galactica for molecule generation without the retrieved proteins by BioBridge, as the baseline, to Table 21, on Page 21
- Add the training setup description to Appendix I, on Page 23

Changes in the revision are visually highlighted in red.

---

### Meta-Review · Area_Chair_BQFU · 2023-12-19

**Metareview:**

This paper proposes BioBridge, a method to train a joint foundation model for protein, molecule, and clinical data. BioBridge's core learning approach is based on contrastive learning. The resulting model demonstrates strong performance relative to several baseline knowledge graph (KG) embedding methods on multiple tasks.

Strength of the paper:
1. the paper proposes a contrastive learning method to integrate different types of nodes in one unified framework.
2. The experiments are extensive in validating the performance of the proposed algorithm.

Weakness of the paper:
1. The drug molecule generation setup misses quantitative evaluation of the model's capability in generating functional and novel drug molecules. It is better to either remove this part or adding more evidence.
2. The baselines on graph embedding methods are out dated.

The additional theorem added in the paper seems obvious and it does not seem important to the main method.

All reviewers are satisfied with the contribution in the paper and author responses.

**Justification For Why Not Higher Score:**

The paper still leaves some doubt on molecule generation and baselines.

**Justification For Why Not Lower Score:**

The paper is written clear. The method is simple and demonstrated effective.

---

### Decision · Program_Chairs · 2024-01-16

Accept (poster)